# scPRINT: pre-training on 50 million cells allows robust gene network predictions

Jérémie Kalfon [1], Jules Samaran [1], Gabriel Peyré [2] & Laura Cantini [1] ✉

A cell is governed by the interaction of myriads of macromolecules. Inferring such a network of interactions has remained an elusive milestone in cellular biology. Building on recent advances in large foundation models and their ability to learn without supervision, we present scPRINT, a large cell model for the inference of gene networks pre-trained on more than 50 million cells from the cellxgene database. Using innovative pretraining tasks and model architecture, scPRINT pushes large transformer models towards more interpretability and usability when uncovering the complex biology of the cell. Based on our atlas-level benchmarks, scPRINT demonstrates superior performance in gene network inference to the state of the art, as well as competitive zero-shot abilities in denoising, batch effect correction, and cell label prediction. On an atlas of benign prostatic hyperplasia, scPRINT highlights the profound connections between ion exchange, senescence, and chronic inflammation.

Understanding the cellular mechanism is considered a milestone in biology, allowing us to predict cell behavior and the impact of drugs and gene knock-outs[1–5]. A cell is regulated by a complex interplay of myriads of macromolecules that define its state. We can simplify these interactions via a gene network[6] (GN). Many approaches have been developed to infer these networks, focusing on transcription factor (TF)-to-gene links using single-cell omics data modalities like scRNA-seq and scATACseq[7–19]. This gene network subset regulating the cell gene expression levels is often called a gene regulatory network (GRN). However, many other gene products than TFs impact RNA abundances in the cell, like RNA-RNA and protein-TF interactions[20–25]. Most GRN inference methods do not scale to the number of genes and cells present in single-cell RNA datasets, and they need many cells, thus impairing their ability to reconstruct cell-state-specific networks. Other methods consider datasets where differentiating cells can be ordered temporally to predict more causal GRNs. While this approach is interesting, temporal ordering is often hard to predict[15,26].

Benchmarks like BeeLine[27] and MCalla et al.[28] have shown that despite the existence of many methods, GN inference remains a challenging problem. Indeed, it is underconstrained and has limited prior knowledge. New foundational models trained on tens of millions of measurements could help solve these difficulties. Transformers like BERT[29,30] have gained traction in computational biology and have held

promise to learn a model of the cell that would translate across many tasks of cellular biology, such as cell type annotation, batch-effect correction, perturbation prediction, and GN inference[31]. Among them, scGPT[32] got much attention, proposing a novel encoding of genes and their expression, a new pretraining methodology similar to autoregressive pretraining in language models, and the possibility of extracting GRNs (see methods).

Inspired by these efforts, we propose scPRINT, a foundation model designed for GN inference. scPRINT brings inductive biases and pretraining strategies better suited to GN inference while answering issues in current models (see Table S1). scPrint outputs cell type-specific genome-wide gene networks but also generates predictions on many related tasks, such as cell annotations, batch effect correction, and denoising, without fine-tuning.

We extensively benchmark scPRINT on challenging GN inference tasks, from literature-based networks to cell type-specific ones generated via orthogonal sequencing methods. We show that scPRINT outperforms the state of the art on most of these atlas-level benchmarks. In addition, our model focused on GN inference, is also competitive on a compendium of tasks like denoising, cell type prediction, and embedding with batch effect correction. This suggests that by learning a cell model, scPRINT gains zero-shot abilities in many tasks of cellular biology.

[1]Institut Pasteur, Université Paris Cité, CNRS UMR 3738, Machine Learning for Integrative Genomics group, F-75015 Paris, France. [2]CNRS and DMA de l'Ecole Normale Supérieure, CNRS, Ecole Normale Supérieure, Université PSL, 75005 Paris, France. ✉e-mail: laura.cantini@pasteur.fr

We use scPRINT to analyze an atlas of normal and senescent prostate tissues where we identify rare cell populations with early markers of the tumor microenvironment in B-cells. In fibroblasts, we study GNs and recover known hubs such as PAGE4, linking the senescence of fibroblasts to changes in the ECM and downstream inflammation. We find key interconnected pathways of the oxidative stress response and extracellular matrix building via metal and ion exchange in the gene network of BPH-associated fibroblasts. We also show that healthy and disease-related cells exhibit different network patterns, demonstrating that scPRINT can help identify novel pathways and targets while considering them in their specific cellular and molecular contexts.

scPrint[33] (https://github.com/cantinilab/scPRINT) is a fast and open-source tool that can be readily integrated into the bioinformatics pipeline. We make public the code and model weights, but also the pretraining strategies, datasets, and our own dataloader for use with vast training sets like the cellxgene database[34]. We also release a Gene Network benchmarking suite: BenGRN[35] and GrnnData[36].

## Results

### scPRINT: a scRNAseq foundation model for gene network inference

We propose scPRINT (single-cell PRe-trained Inference of Networks with Transformers, Fig. 1A), a state-of-the-art bidirectional transformer designed for cell-specific gene network (GN) inference at the scale of the genome. scPRINT is trained with a custom weighted-random-sampling method[37] over 50 million cells from the cellxgene[34] database from multiple species, diseases, and ethnicities, representing around 80 billion tokens (see Methods). We train scPRINT at various scales

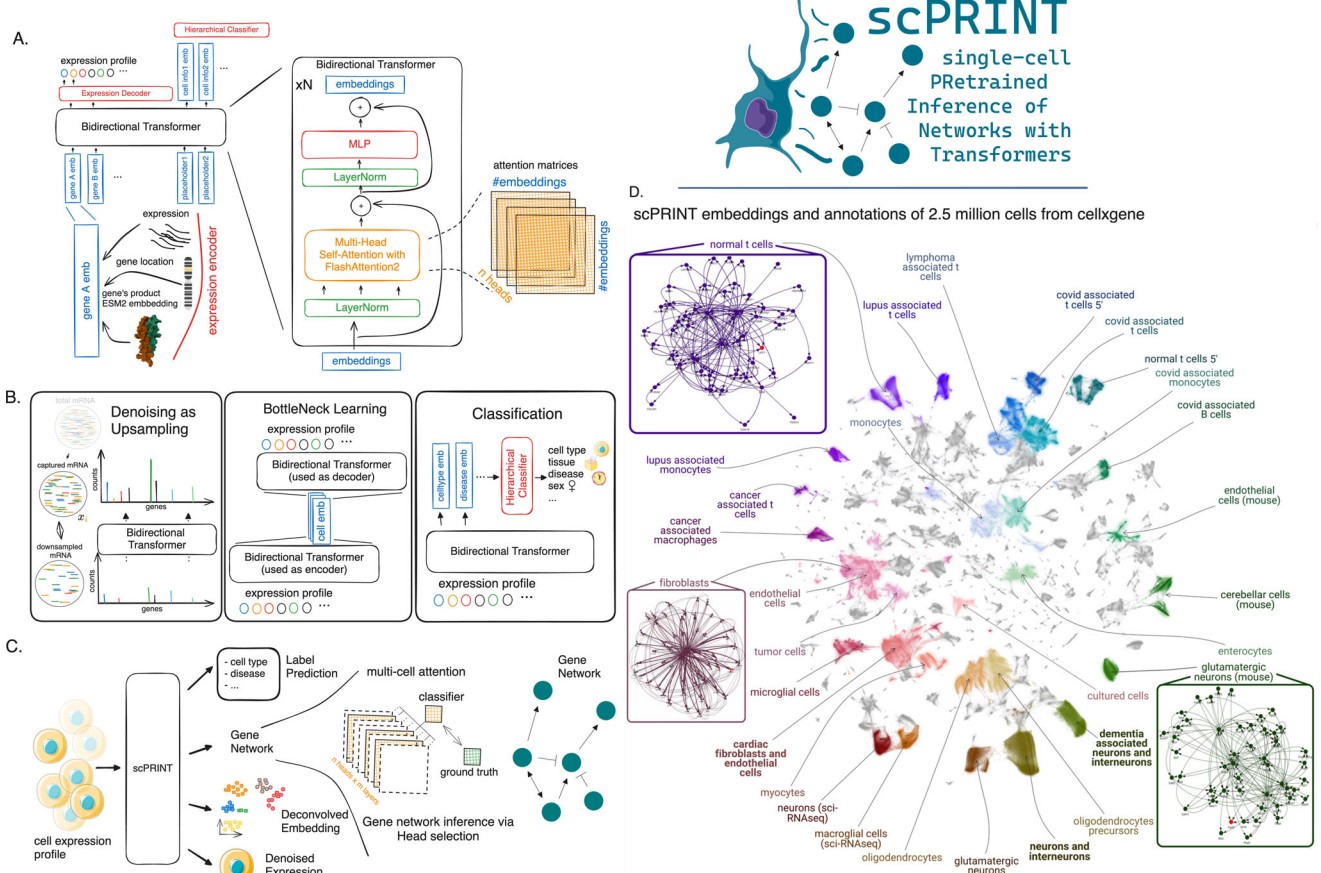

**Fig. 1 | presentation of the scPRINT model and training. A** Schematic representation of scPRINT with its bidirectional encoder, gene expression embedding, gene location encoding, matched ESM2 protein embedding, and gene expression decoder. **B** scPRINT pre-training tasks: denoising task whose goal is to recover the known transcriptomic profile from a purposefully downsampled expression profile. Bottleneck learning reconstructs the expression of requested genes using only their cell embedding. The same model is used for both The encoding and decoding steps. Classification is achieved by applying a hierarchical classifier to each disentangled embedding. This pushes the first embedding to contain cell type information, the second embedding to contain disease info, and so on (see methods). **C** The different outputs in scPRINT. scPRINT generates label predictions of cell type, tissue, disease, sex, sequencer, ethnicity, and organism. scPRINT generates multiple embeddings (which we call disentangled embedding), a general one, as well as a specific embedding for each class. scPRINT also generates a reconstructed expression profile at any requested sequencing depth (i.e., total transcript count). scPRINT also generates a Gene Network by selecting and combining various attention heads into a gene by gene matrix. **D** Example of a scPRINT output from a random subset of 2.5 million cells from the cellxgene database.

Embeddings and labels are generated by scPRINT, together with the example cell type-specific gene networks. We show only subparts of the networks extracted from a central node, represented in red. **A** chromosome_3 icon by Helicase 11 undefined is licensed under CC-BY 4.0 Unported https://creativecommons.org/licenses/by/4.0/. **B** fat-tissue icon by Servier https://smart.servier.com/ is licensed under CC-BY 3.0 Unported https://creativecommons.org/licenses/by/3.0/. tumor icon by Servier https://smart.servier.com/ is licensed under CC-BY 3.0 Unported https://creativecommons.org/licenses/by/3.0/. **B**, **C** normal-cell-2 icon by Servier https://smart.servier.com/ is licensed under CC-BY 3.0 Unported https://creativecommons.org/licenses/by/3.0/. **D** The scPRINT logo contains a modified version of: langerhans-cell icon by Servier https://smart.servier.com/ is licensed under CC-BY 3.0 Unported https://creativecommons.org/licenses/by/3.0/. https://doi.org/10.2210/pdb1nkp/pdb initially in Nair, S.K., Burley, S.K. "X-ray structures of Myc-Max and Mad-Max recognizing DNA: Molecular bases of regulation by proto-oncogenic transcription factors", (2003) Cell 112: 193-205 visualized with mol*. Licensed under CC-1.0 Universal https://creativecommons.org/publicdomain/zero/1.0/.

(from 2 M to 100 M parameters) and very efficiently by using flashattention2[38], e.g., only requiring an A40 GPU for 48 h to train our medium model, significantly reducing the barrier to entry for any computational biology lab (see Table S2).

To push scPRINT to learn meaningful gene networks (GN) and an underlying cell model, we design a unique set of pretraining tasks, as well as expression encoding and decoding schemes (Fig. 1B).

scPRINT's pretraining is composed of three tasks which loss are added and optimized together: a denoising task, a bottleneck learning task, and a label prediction task. The objective is to let scPRINT learn to represent meaningful gene connections while also endowing it with a breadth of zero-shot prediction abilities.

Indeed, similarly to ADImpute[39,40], we expect a good GN to help denoise an expression profile by leveraging a sparse and reliable set of known gene-gene interactions.

We implement this denoising task as the upsampling of transcript counts per cell (see Methods). While most other methods have used masking as a pretraining task, our method is related to the downsampling and masking task of scFoundation[41]. We show that this strategy performs better than masked language modeling and gives scPRINT the ability to upsample any expression profile.

In addition, we expect that a cell model tasked to compress expression profiles into embeddings can learn the regularities of modules and communities of gene networks. Therefore, the bottleneck learning task drives scPRINT to generate an embedding and a cell expression profile from its embedding only. The embedding is generated by scPRINT and is used again, without any cell expression value, to regenerate the true expression profile (see Methods).

Finally, the cell's gene network should represent the cell state and its different phenotypic facets. Effectively, scPRINT generates not just one embedding per cell but multiple. A hierarchical classifier is then applied to each distinct cell embedding to predict its associated class, such as cell type, disease, sex, organism, ethnicity, and sequencing platform. The embeddings thus become disentangled, each representing a specific facet of the cell state[42]. This last training task pushes the large cell model and its gene network to represent the cell state[42].

Thanks to the cellxgene database requirement for complete annotations and our innovative hierarchical classifier, we have added label prediction as part of the pretraining of scPRINT. While the assumption is that in other modalities, the scarcity and noisiness of such labels make it infeasible, we show that this approach is a net positive in our case (see Table S3, Methods). Indeed, it helps us disentangle the various cell embeddings and performs zero-shot predictions on unseen datasets. These disentangled embeddings are opening a future possibility to perform counterfactual generation: mixing embeddings representing different facets of cell states, e.g., fibroblast + cancer + pancreas tissue + female, to generate novel unseen expression profiles.

scPRINT converts the gene expression of a cell to an embedding by summing three representations or tokens: its id, expression, and genomic location (Fig. 1A, see Methods). scPRINT encodes the gene IDs using protein embeddings. This gene representation is made using the ESM2[43] amino-acid embedding of its most common protein product (see Supp Fig. S1). First proposed in UCE[44], the model learns to leverage representations that can potentially apply to unseen genes and species, using the structural and evolutionary conservation of the sequence encoded by ESM2. While drastically reducing the number of weights trained for the model compared to scGPT and Geneformer (see Methods), this representation also contains some priors needed to infer protein-protein[45] interactions (Fig. 1A).

The gene's expression is tokenized via a multi-layer perceptron (MLP) using log-normalized counts. This MLP lets the model learn a metric behind gene expression, whereas scGPT and Geneformer apply a specific prior for the encoding of their gene expression (see Methods).

Finally, we help the model know that genes with similar locations tend to be regulated by identical DNA regions, using the positional encoding of their location in the genome (see Methods).

These three embeddings are summed and then concatenated across the genes expressed in a cell together with additional placeholder cell embeddings to form the transformer model's input matrix.

scPRINT is pretrained using 2200 randomly selected expressed genes in a cell profile. If a cell doesn't have enough expressed genes, the list is padded with randomly selected unexpressed genes. A context of 2200 genes, while not genome-wide, captures all the expressed genes in more than 80% of the cell profiles in the cellxgene database. We also show that scPRINT can make predictions on much larger sequences of genes at inference time without using attention approximation methods[46].

Using unexpressed genes, combined with the denoising task, let scPRINT discriminate the true zeros from dropouts in scRNAseq[47]. The expression decoder of scPRINT further helps model this statistic of the data. It is a zero-inflated negative binomial graphical model inspired by previous literature in single-cell RNAseq modeling[48]. Here, the loss (also used for bottleneck learning) is thus the negative log-likelihood of the gene expression given the distribution parameters.

As shown in Fig. 1C, at inference time, scPRINT can generate multiple outputs across any scRNA-seq-like cellular profile of various mammalian species without the need for fine-tuning. Figure 1D shows scPRINT's prediction at the scale of an atlas of 2 M randomly sampled cells from cellxgene. A critical emergent output of scPRINT is its cell-specific gene networks. Following a similar approach to ESM2, we generate cell-level gene networks via the bidirectional transformer's input-wise weighted matrices, called attention matrices -or heads-. They represent general gene-gene connections and can be subsetted to TF-gene connections (i.e., GRNs). Remarkably, we made this approach scalable enough to compute attention heads-based gene networks for 1 to 10,000 cells, at the genome scale, with commodity hardware and in a few minutes. These networks both showcase the ability of scPRINT to model cellular biology and help make it a more explainable tool for the community, showing the network assumptions made during inference. The attention heads are either all aggregated by averaging or can be selected to better reflect connections of interest (Fig. 1C). This is done using the average of the heads most correlated with literature or perturbation-based ground truth networks. Finally, while we do not assess scPRINT's ability to model inhibition due to the scarcity of such annotations, we leave open the possibility of using our head selection technique for such a task.

Similarly to what has already been done in ESM2 and the Large Language Model literature[49–51], we deeply investigate the meaning of attention matrices in the context of cellular biology, an aspect understudied in the literature of foundation models applied to genomics.

In the following sections, we benchmark scPRINT on gene network inference against scGPT, DeepSEM[19], GENIE3, and Geneformer v2[52], the updated version of Geneformer. scGPT and Geneformer v2 are highly cited and published transformer models for single-cell RNAseq, mentioning the inference of gene interactions[32]. DeepSEM is an autoencoder model jointly learning its weights and a gene network matrix. GENIE3[53] generates networks via regression by finding the set of genes that best predict another gene's expression. It is one of the top-performing and most used methods for GRN inference (see Methods). However, it suffers from very long run times and high memory requirements (see Table S4).

## scPRINT recovers biological features in its gene networks

We benchmark scPRINT against the state-of-the-art based on whether their recovered networks contain meaningful biological knowledge. We consider two main benchmarking methodologies, one using a simulated expression profile from a well-established biological network. Because simulated data does not represent real cell expression data[54] (see Methods), our second and main approach focuses on biological features of a network inferred from real cell expression profiles.

Indeed, we assume that a meaningful gene network should have some of its hub nodes being TFs. TFs should be more connected to their known target, on average. We should recover known gene-gene connections and expect enrichment of cell type-specific marker genes in the network.

We compare each gene network inference method's ability to recover a known network from 1000 simulated single-cell RNAseq expression profiles generated by the Sergio ODE model[54] from the ground truth GN Regnetwork[55] (see Methods). Only scPRINT was able to recover meaningful connections (Table S5). One explanation is that through its training, scPRINT has learned the common gene connections that also exist in the RegNetwork ground truth.

On GN inference from real expression data, we noticed that depending on cell type and datasets, the different tools could vary greatly in the similarity of their GNs to the Omnipath[56] ground truth. Because of this, we focused our benchmark on three randomly selected test datasets of kidney, retina, and colon tissues comprising 26 cell types[57–59] (see Methods, per dataset results in Supp. Fig. S2). Of note is that we could not determine if these datasets were used during the training of scGPT or Geneformer.

We predict one network per cell type, using the same 1024 cells and their 5000 most differentially expressed genes for all benchmarked methods. We evaluate the quality of the networks based on their overlap with Omnipath. We also compute the network's enrichment for cell type markers, TFs, and ENCODE TF targets[60] using the prerank[61] algorithm (Fig. 2A).

Although the scGPT code mentions GRN inference only using perturb-seq data, we reapply the same method without the perturbation-baseline comparison. This is to make it comparable with other benchmarked methods and because most of our datasets are not perturbation-based. Similar to what is presented in its paper, we use the mean of the attention matrices across cells and the four attention heads of the last layer of the human pre-trained model. We retain this method across our benchmarks for scGPT (see Methods). We apply a similar strategy for Geneformer (see Methods).

For scPRINT, we generate three network versions: one simply called *scPRINT*, based on the average of all heads in the model. *scPRINT (omnipath's heads)*, based on the average of heads selected with our abovementioned head selection method inspired by ESM2, and *scPRINT (genome)*, which is like the *scPRINT* network but uses our method to generate genome-wide networks (see Methods) instead of using the 5000 most differentially expressed genes. Indeed, in transformer models, the choice of attention heads is important. Although transformers can learn the causal structure of their input, it has been shown that some attention heads, especially in larger networks, can become unused, containing predominantly random connections[62]. Some work has been done at pruning these heads[63] or forcing a head selection mechanism at inference and training[64]. For *scPRINT (omnipath's heads)*, we select heads based on a linear classifier's prediction of the best set of heads to predict a subset of Omnipath (see Methods). Similarly to the *scPRINT* network, these heads are then averaged to generate the *scPRINT (omnipath's heads)* gene network. To perform this selection, we split the omnipath dataset into train/test and select heads, using 50% of the ground truth and only the first cell type of each dataset. We then use the same combination of heads across all other cell types. This shows that our selection process builds consistent networks across cell types and parts of the ground truth. This approach contrasts with previous ones like scGPT's and GENIE3 by using part of an available ground truth to select heads.

First, we look at how much information from Omnipath is contained in the inferred networks. Omnipath[56] contains around 90,000 curated gene-gene connections, mainly from the literature. These connections are cell type agnostic, and most are TF - gene. On this benchmark, we evaluate the networks based on Area Under the Precision-Recall Curve (AUPRC) and Early Precision Rratio (EPR), two metrics often used in GRN benchmarks[27] (see Methods), where we define our task as a binary classification of connections on all gene-gene pairs. Due to the row-wise normalization of networks generated by all methods, and because Omnipath has many sources with only a few targets (see Supp Fig. 2), we here use the transpose of our inferred networks when making comparisons with Omnipath (see Methods).

In Fig. 2B, we can see that *scPRINT (omnipath's heads)* outperforms all methods on average across all cell types. While *scPRINT (omnipath's heads)* uses some ground truth information to select its head, we see that *scPRINT* still outperforms scGPT and Geneformer v2 on the EPR metric, showing that its top predicted edges more closely match the ground truth.

AUPRC results are very low overall because we do not expect most Omnipath connections to be present in the cell type's gene network, as many connections in Omnipath might only be true in some cellular contexts. Moreover, we do not expect most connections in our generated network to exist in Omnipath as it only contains a small fraction of all real gene-gene connections. Although overall AUPRC values are small, we can see that both scGPT and scPRINT outperform the other methods in the number of connections recovered. Indeed, on average, scGPT and scPRINT respectively recover 42% and 67% more connections than GENIE3.

However, GENIE3 is often used by biasing the method to only predict TF-gene connections (see Methods). This type of network, usually called a gene regulatory network (GRN), is most often used, given the importance of TFs in regulating gene expression. To compare the other methods to this GRN version of *GENIE3*, we also use a GRN version of their networks by subsetting them to TF-gene connections only. In this context, all the methods significantly improve their predictions without altering their relative performances (Fig. 2B). This is unsurprising, considering that Omnipath is strongly biased towards TF-gene interactions.

Interestingly, we have seen that smaller scPRINT models containing fewer heads perform better when taking the average of their heads. In contrast, head selection is often more advantageous in larger models with more heads (see Table S6). As presented at the beginning of the results section, it might be that as models become larger and less regularized, some heads tend to become unused and contain mostly noise. As a consequence, a head selection is advantageous in larger models.

We also expect biologically meaningful gene networks to have their central nodes enriched for TFs. In addition, because these networks are cell type-specific, we expect their central nodes to be enriched for some marker genes of their associated cell types (see Methods). In this regard, both scGPT and scPRINT achieve very similar and strong network enrichment for TFs compared to GENIE3, DeepSEM, and Geneformer v2, whose networks are not enriched for TFs (Fig. 2C). Moreover, amongst the 178 cell types we have marker gene sets for in pangaloDB[65], all methods find some enrichments, especially GENIE3 and scGPT (see Methods). We notice that selecting heads based on Omnipath significantly improves scPRINT's network enrichment for cell-type markers. Of note, our goal is not to annotate cell types from the gene network but mainly to showcase the network's cell type specificity.

Finally, we also examine how much the connections of each TF are enriched for that TF's target. Here, scPRINT overperforms all other methods (Fig. 2D). In the *scPRINT* networks, 20% of the Transcription Factors for which we have data on ENCODE have connections significantly enriched for their ENCODE-validated gene targets[66]. Interestingly, only our large cell model achieved a great performance, and scGPT did not display any enrichment across the 26 cell types assessed. While we acknowledge that ENCODE is used in the Omnipath database, we cannot expect Omnipath to represent the ENCODE targets. Indeed, it combines and processes 57 additional data sources to build its consensus network.

*scPRINT (genome)* has been added despite its performance not being comparable to other methods. Indeed, comparing its overlap

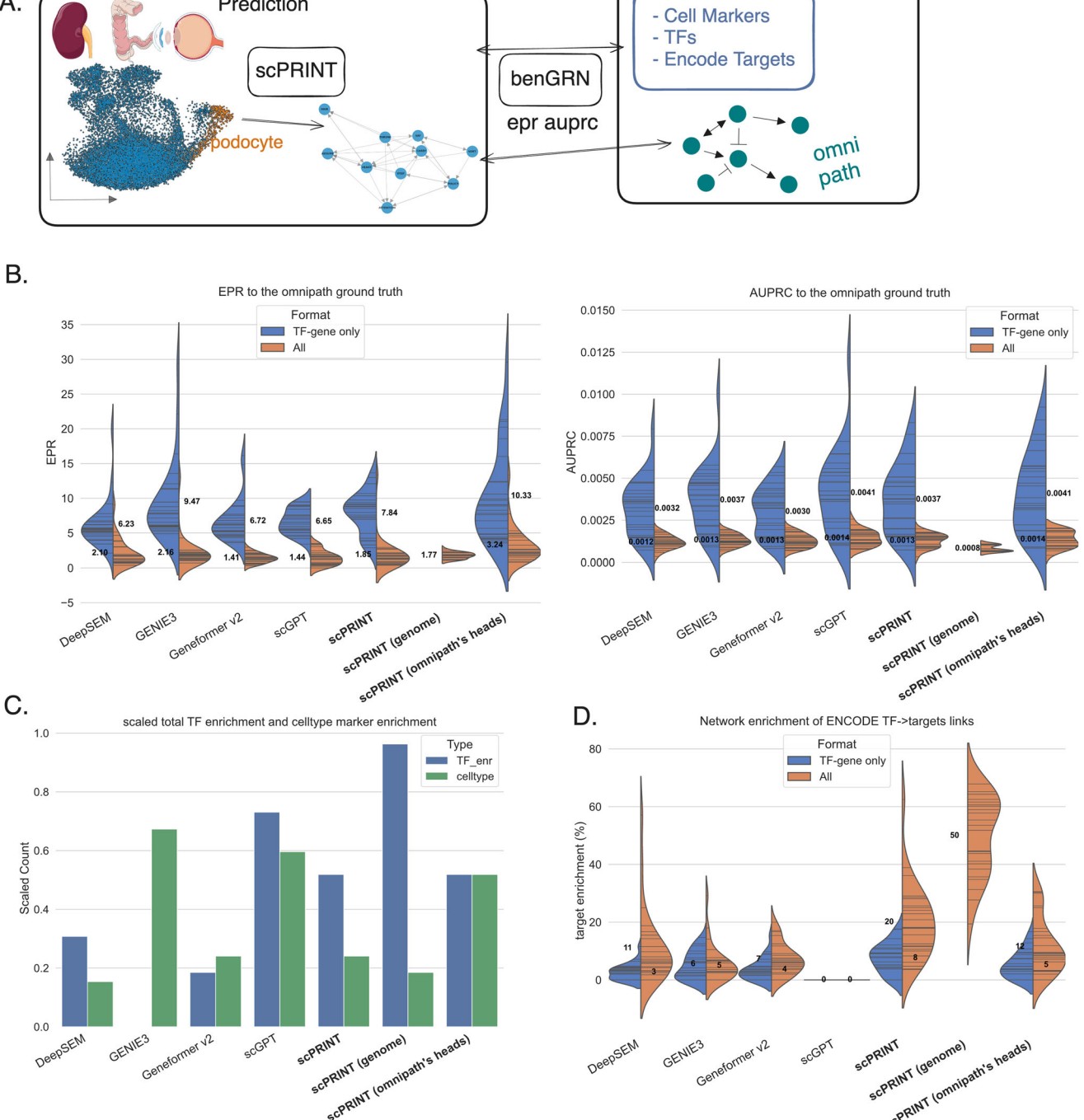

**Fig. 2 | Analysis of the gene networks generated by scPRINT. A** We extract cell type-specific gene networks for each cell type in the dataset ($n = 26$ cell types across 3 datasets). We perform Gene Set Enrichment Analysis (GSEA)[61] on the network's nodes ($n = 4000$ genes). We compute the ability of the edges to recover the Omnipath ground truth's connections. **B** Violin plot of the ten different Area Under the Precision Recall Curve (AUPRC) and Early Precision Rratio (EPR) values obtained when comparing the inferred cell type-specific networks with the Omnipath network for scPRINT: average of all attention heads, scPRINT (genome): same scPRINT version but computing a genome-wide gene network, scPRINT (omnipath's heads): same scPRINT version but with attention heads selected using a subset of omnipath, scGPT, DeepSEM,

Geneformer v2, and GENIE3, when considering only Transcription Factor (TF)-gene connection or all gene-gene connections. **C** Violin plot of the average number of TF with enrichment for their ENCODE target in each cell-type-specific network. **D** Number of GNs with a significant enrichment of TFs and of their cell type's marker genes. Source data are provided as a Source Data file. **A** kidney-2 icon by Servier https://smart.servier.com/ is licensed under CC-BY 3.0 Unported https://creativecommons.org/licenses/by/3.0/. healthy-colon-3d icon by Servier https://smart.servier.com/ is licensed under CC-BY 3.0 Unported https://creativecommons.org/licenses/by/3.0/. eye-exploded-view icon by Servier https://smart.servier.com/ is licensed under CC-BY 3.0 Unported https://creativecommons.org/licenses/by/3.0/.

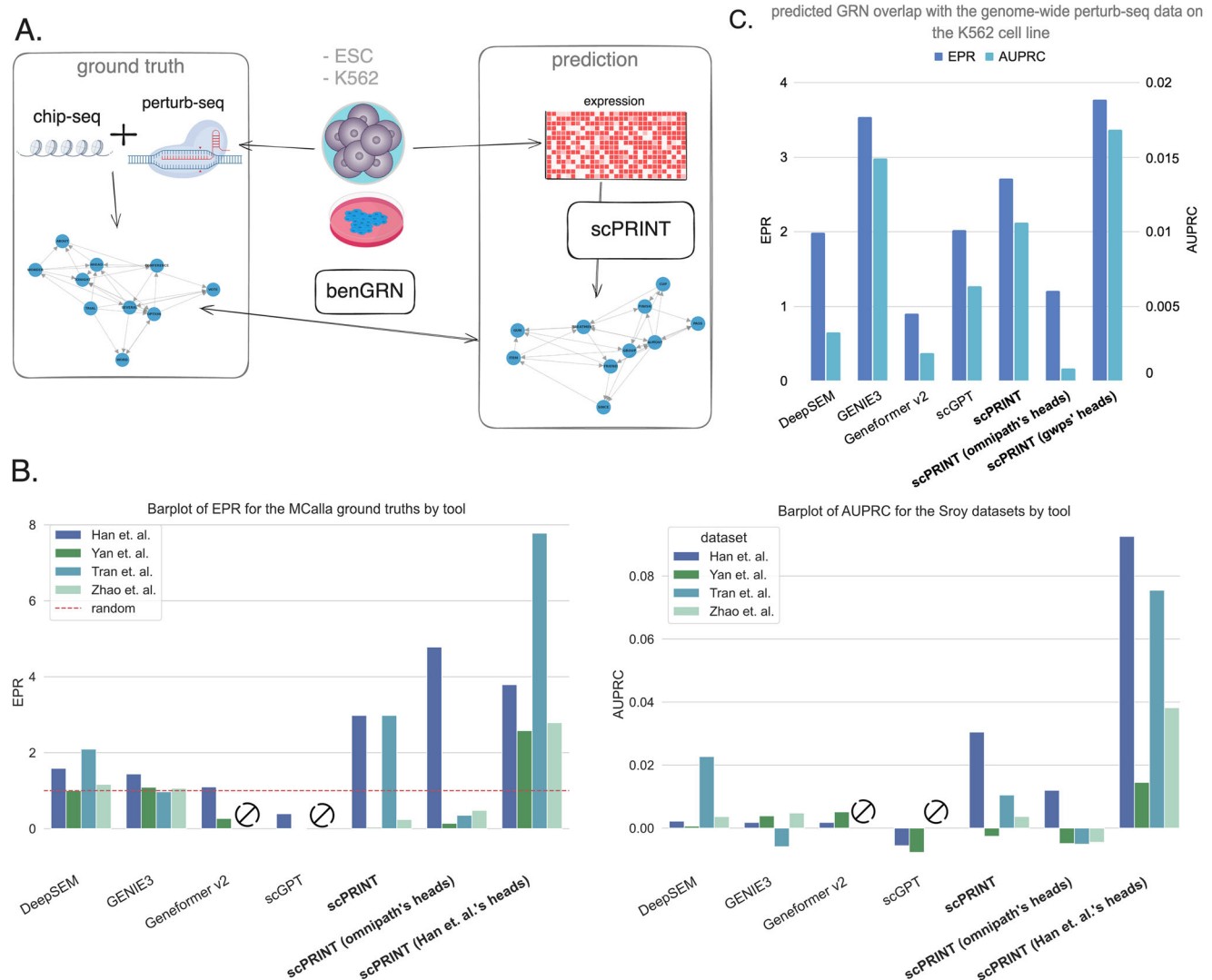

**Fig. 3 | scPRINT GN inference performance on cell-type specific ground truths.**
**A** The ground truths are generated via orthogonal sequencing assays on the same cell type. ChIP-seq and perturb-seq are intersected for the MCalla et al. dataset on human (hESCs) and mouse (mESCs) Embryonic Stem Cells, whereas perturb-seq on the K562 cell line is only used for the genome-wide perturb-seq ground truth.
**B** Performance of scPRINT, scPRINT (omnipath's heads): same scPRINT version but with attention heads selected using a subset of omnipath, scPRINT (Han al.'s heads): same scPRINT version but with attention heads selected using a subset of

the Han et al.'s ground truth dataset, compared to GENIE3, DeepSEM, Geneformer v2, and scGPT on the MCalla et al. ground truth using the AUPRC and EPR on two human and two mouse ESC datasets. **C** Same as (**B**) but on the genome-wide perturb-seq dataset (with scPRINT (Han et. al.'s heads) replaced with scPRINT (gwps' heads): same scPRINT version but with attention heads selected using a subset of the genome-wide perturb-seq ground truth). Early Precision Ratio (EPR) and Area Under the Precision-Recall Curve (AUPRC) are provided here in one barplot, left to right. Source data are provided as a Source Data file.

with Omnipath is unfair as it includes many more genes and connections, many of which will have almost no data on this ground truth. While *scPRINT (genome)* showcases our ability to generate genome-wide networks, it also shows strong performances in TF enrichment and ENCODE TF-target enrichments. This highlights that even at such a large scale, networks generated by scPRINT are enriched in biological knowledge gained solely from its pre-training tasks.

Overall, we have shown that scPRINT generates, in one forward pass, cell type-specific gene networks that are biologically meaningful. We will now examine them using cell type-specific ground truths extracted from orthogonal experiments.

### scPRINT outperforms the state of the art on cell type-specific ground truths

Although we have shown that our networks represent meaningful biology, the Omnipath ground truth is literature-based and not cell type-specific. Here, we use two different modalities, perturb-seq[67], and

ChIP-seq[68], as ground truths to compare predicted gene networks against.

In the MCalla et al.[28] ground truth, ChIP-sequencing and perturb-seq are intersected to get at the small subset of possibly direct connections between TFs and genes for both human and mouse embryonic stem cells (ESC) (Fig. 3A, see Methods). We have seen that these ground truth networks show a different pattern than literature-based networks (see Supp Fig. S3). Some TFs regulate only a few genes, whereas others are highly connected.

To generate our networks, we use as input one human and two mouse ESC scRNA-seq datasets from MCalla et al. with the addition of another human dataset from Yan et al.[69] Networks are generated over the same 1024 cells, and the 5000 most variable genes for all methods. For scPRINT, three networks have been generated: one averaging all the attention heads (*scPRINT*), one averaging heads selected based on how well they predicted Omnipath ground truth data: *scPRINT (omnipath's heads)*, and one averaging heads selected from one of the MCalla

ground truths: *scPRINT (Han* et al.*'s heads)*. For more details, see the results section 2: scPRINT recovers biological features in its gene networks. Of note, due to the small amount of genes assessed in the ground truth, we do not add the genome-wide network version here. Moreover, only the TF version of *GENIE3* and the TF-gene subsets of the other method's networks are used since the ground truth only contains TF-gene connections.

Contrary to Omnipath, some elements in these biological networks are highly connected, whereas many others display no connections. This imbalance means that a method predicting only the highly connected TFs will perform well on the MCalla et al. benchmark. As a consequence, we are not transposing the attention matrix as done in the previous section.

Based on both AUPRC and EPR, scPRINT outperforms all other methods on this benchmark (Fig. 3B). This means, for example, that when training GENIE3 to only predict a gene's expression based on TF expressions, it is not selecting the right TFs amongst the set of a few dozen assessed in MCalla et al.

scGPT, Geneformer v2—and, in a few cases, scPRINT—can have values worse than random guessing. Thus, their predictions are often specific to some TFs but not necessarily the right ones (Fig. 3B).

It also appeared that selecting heads based on Omnipath, although helping slightly in one instance, is not a net benefit for this dataset (see Table S9). This makes sense since MCalla et al. itself does not overlap much with Omnipath (see Table S7). However, selecting heads based on the ground truth itself, only using 50% of the connections available, shows substantial improvement. These same heads also show reliable behavior when using them on the second dataset and ground truth of the same species.

This shows that scPRINT can better decipher direct from indirect TF-gene connections than scGPT, DeepSEM, Geneformer v2, and GENIE3, although more tests would likely be needed. However, the results also highlight that the high data imbalance (i.e., TFs being not connected or highly connected) combined with the small dataset size (i.e., only a few dozen TFs assessed) and the low number of cells make the results in MCalla et al. very variable. Some of this might be true biology or explained by ChIP-seq, which can be very noisy depending on the quality of its antibodies[70].

To answer this issue, we selected another dataset: genome-wide perturb-seq (gwps)[71]. Here, we measured the effect on transcription of knocking out all expressed genes in the K562 cell line. We transformed it into a network using a cutoff of 0.05 on the significance level of each gene's differential expression before and after the Knock Out (KO) of each other gene. Although this does not tell us which connections are direct or indirect, we now have a much broader set of connections over thousands of genes and better statistics to assess our gene network inference methods.

GENIE3 performs best, directly followed by scPRINT. Interestingly, Geneformer v2 shows poor performance (Fig. 3C). Perturbation experiments are known to correlate somewhat to expression correlation, and this might explain GENIE3's strong performance. However, when using our head selection mechanism, *scPRINT (gwps' heads)* outperforms GENIE3. Again in this dataset, selecting heads based on Omnipath does not help; the small overlap between the gwps network and the Omnipath ground truth network seems likely to be the culprit (see Table S7). These overlaps show that the three ground truth networks are very different and that a different set of heads predicts each type of ground truth. We also assess the networks on the TF-gene only subset of the gwps ground truth. Here, we see a large drop in performances for most methods, except GENIE3 (see Supp. Fig. S4).

Finally, we have seen that on both MCalla and gwps, scPRINT also predicts networks that agree with the Omnipath ground truth and are again enriched for cell type markers and TFs (see Tables S8, S9).

Since GNs can be seen as approximations of a cell model, we expect that when a tool has good internal cell models, it should generate meaningful results on tasks such as denoising, cell type prediction, embedding and batch effect correction, perturbation prediction, trajectory inference, and more. We will now focus on those three tasks, which are orthogonal to GN inference to compare the ability of scPRINT to the state-of-the-art.

## scPRINT is competitive on tasks orthogonal to GN inference

To test the quality of the cell model learned by scPRINT, we now consider denoising, cell type prediction, and batch effect correction as a representative set of classic scRNAseq and cellular biology benchmarks.

Similarly to our pretraining task, we simulate lower transcript count profiles and then ask scPRINT and two other state-of-the-art methods, MAGIC[72] and KNNsmoothing2[73], to recreate the true expression profile. We use Spearman correlation to the original gene expression profile as our metric. In Fig. 4A, we show the increase in correlation after denoising the downsampled profile on 3 test set datasets, composed of ciliary body, colon, and retina tissues[58,74,75], randomly selected from cellxgene (see Methods).

scPRINT is competitive with both SOTA methods, while contrary to MAGIC and KNNsmoothing2, it operates independently over each cell in the test set (see Methods). We have also seen a 10% variability in denoising ability across the different datasets used (see Table S10). This was similar across all tools and possibly related to the number of genes expressed in each dataset. However, these test cases mostly contain very similar cell states, whereas denoising is helpful in cases with rare cell types or transitory cell states that have low cell counts by default. We show that since scPRINT does not aggregate profiles over neighboring cells, it outperforms MAGIC and KNNsmoothing2 in rare cell states subsets of the datasets (respectively: pericytes, microfold cells of epithelium of small intestine and microglial cells) with around 10 to 200 cells (Fig. 4A, Supp Fig. S5). Computing MAGIC and KNNsmoothing2 over only this rare cell population gives even lower performances for MAGIC and creates an error for KNNsmoothing2 (see Table S10). These results suggest that a good cell model, that has learned reliable gene-gene interactions, can help denoise an expression profile.

For cell type classification, we expect scPRINT to be able to find sets of genes that can predict a cell type across multiple batches and under the high dropout rate of single-cell RNAseq. To evaluate cell type classification, we use the multi-batch benchmark pancreas dataset of openproblems, its metrics, preprocessing, and hyperparameter choices (see Methods)[76,77].

scPRINT is a zero-shot predictor of cell labels. Indeed, it does not need to train on the dataset itself to make its predictions, unlike other methods that often need to use >70% of the test dataset for training. scPRINT also makes predictions over >200 cell type labels, while other methods often only predict a few cell types. Conversely, the other classifier methods, like Logistic Regression or XGBoost, and previous foundation models are trained or fine-tuned on the test dataset, thus giving a strong advantage over scPRINT. We, therefore, also compare scPRINT to the marker-based classifier CellTypist[78] and its pancreas marker database (see Methods). A method that also does not use the labels of the test dataset.

In this benchmark, scPRINT reaches 62% classification accuracy, largely outperforming CellTypist (Fig. 4B, Supp Fig. S6). Interestingly, with the macro F1 score, which considers each cell type group equally regardless of its size, scPRINT achieves similar results to the state-of-the-art[77] methods: Logistic Regression and XGBoost. This is probably because scPRINT is not influenced by the number of cells in each category.

In addition, we have noticed that scPRINT is challenged by some specific pancreatic cell types in this dataset. Indeed, scPRINT often switches the assignment of A, B, D, and E cells. Thus, when using the coarser "endocrine pancreatic cell" label to define these cell types, we

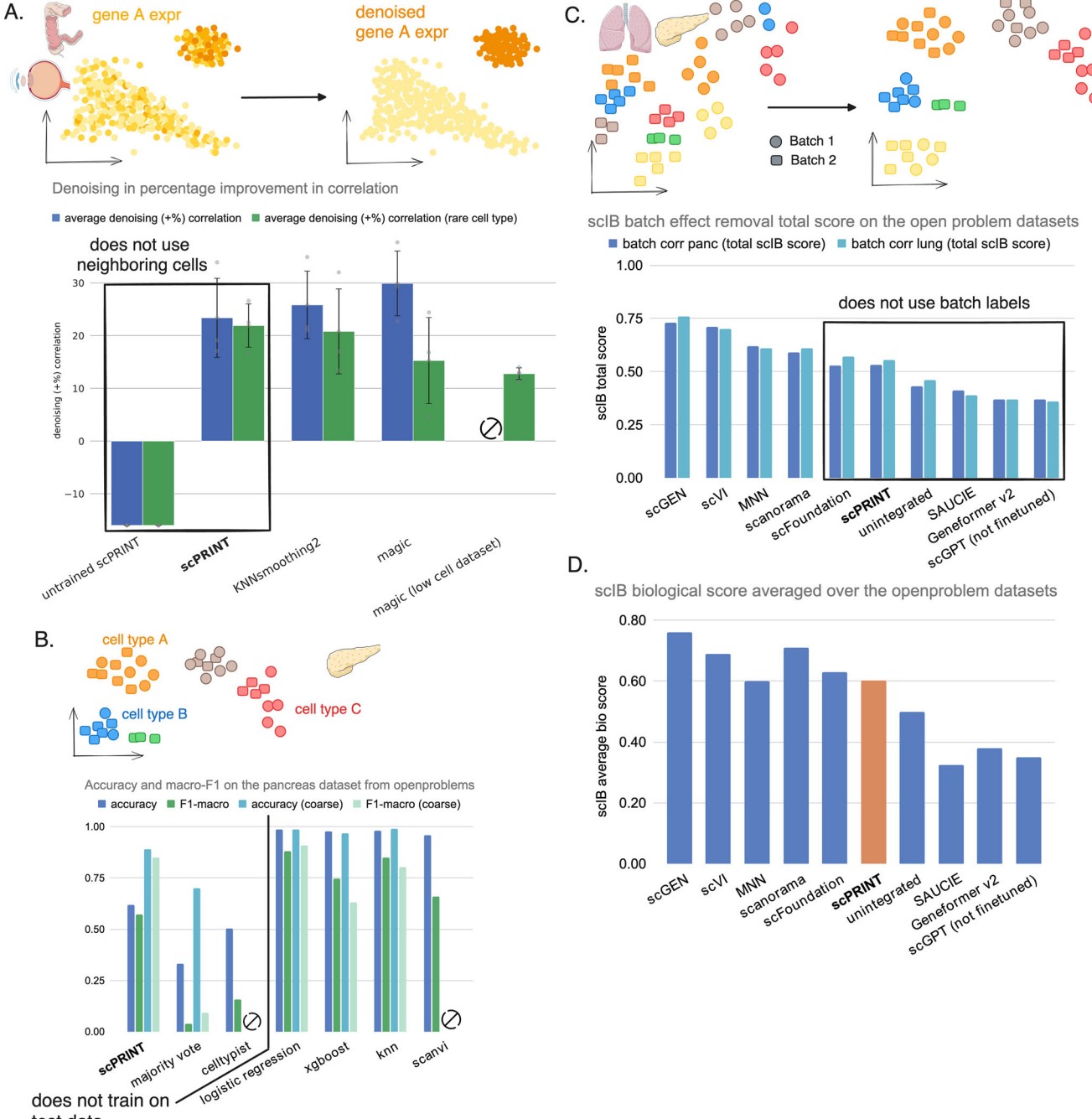

**Fig. 4 | Benchmark of scPRINT on orthogonal tasks to GN inference.**
**A** Performance for the denoising task compared to state-of-the-art methods MAGIC and knnsmooth2 on 3 datasets (ciliary body, colon, and retina tissues) from cellx-gene. Here, we generate a noisy profile by downsampling 70% of the cell transcripts and computing the Spearman correlation increase of the correlation between the denoised and the true profile compared to the one between the noisy and the true profile. **B** Performance on cell-type label prediction compared to state-of-the-art methods as well as CellTypist. Showing accuracy, F1 and macro-F1 scores for the open-problems human pancreas dataset. **C** The performance of scPRINT as well as scGPT and Geneformer v2 on batch effect correction on the human pancreas and lung datasets from the openproblems challenge showing the scIB aggregated

score. They are compared to state-of-the-art methods which results were extracted from the openproblems benchmark. Unintegrated means only PCA was applied. **B** The scIB avgBIO score on both datasets. Source data are provided as a Source Data file. **A** healthy-colon-3d icon by Servier https://smart.servier.com/ is licensed under CC-BY 3.0 Unported https://creativecommons.org/licenses/by/3.0/. eye-exploded-view icon by Servier https://smart.servier.com/ is licensed under CC-BY 3.0 Unported https://creativecommons.org/licenses/by/3.0/. **B, C** pancreas icon by Servier https://smart.servier.com/ is licensed under CC-BY 3.0 Unported https://creativecommons.org/licenses/by/3.0/. **C** lung icon by Servier https://smart.servier.com/ is licensed under CC-BY 3.0 Unported https://creativecommons.org/licenses/by/3.0/.

see a big improvement in the accuracy and macro-F1 score of scPRINT, even outperforming state-of-the-art methods.

Here, we have shown the accuracy of scPRINT independently of cell neighborhood. However, like gene marker-based methods, scPRINT can annotate cell types in novel datasets. In this context,

its predictions could be smoothed and improved using majority voting over predefined cell clusters. Finally, scPRINT predictions are given as probability vector overall cell type labels. They can be used to display the top K labels and learn about the model's uncertainty.

Thanks to its disentangled embeddings, scPRINT can also generate cell representations that partially remove batch effects from cell profiles. On the human pancreas and lung datasets of openproblem[79], we see that, based on the scIB metrics, scPRINT shows convincing batch effects removal ability, while not on par with the SOTA methods scGEN and scVI (Fig. 4C, Supp Fig. S7). Concerning foundation models, scPRINT and scFoundation show strong zero-shot performances compared to Geneformer v2 and scGPT. Except for Geneformer v2, scGPT, and scFoundation, we did not rerun previous algorithms for this benchmark and show their performances from the openproblems portal[77] (open-problems-v2.3.6, march 2024). However, we also ran the Geneformer v2 and scGPT foundation models on the openproblems benchmark and showed that without fine tuning on this specific dataset, they are not able to meaningfully correct for batch effect (see Methods).

Moreover, scPRINT is one of the few methods that do not train on the test dataset and do not use already annotated batch labels. When only looking at methods that do not use batch labels as prior information, e.g. SAUCIE[80], LIGER[81], scPRINT is the top performer. We have also noticed that the scPRINT cell embeddings preserve biological information competitively to state-of-the-art methods (Fig. 4D, Supp Fig. S8). This also exemplifies that a reliable cell model can perform well at disentangling the different facets of a cell expression profile and its underlying batch effect.

Overall, we have seen that scPRINT can achieve zero-shot performances on par with many famous single-cell RNAseq tools on multiple important tasks of single-cell biology, showing that our architecture and pre-training tasks are a powerful new foundation for large cell models.

## scPRINT highlights the role of ion exchange and fibrosis in the ECM of Benign Prostatic Hyperplasia

To showcase the ability of scPRINT, we focus on premalignant neoplasms from an atlas of human prostate tissues[82]. The data contains both normals and pre-cancerous lesions, also called benign prostatic hyperplasia (BPH), across multiple patients. Starting from post-alignment raw counts, scPRINT generates a consistent and batch-corrected embedding of the datasets (Fig. 5A, Supp Fig. S9). scPRINT also annotates the cell type, sequencer, sex, ethnicity, and disease type of each cell with an accuracy of 0.71, 0.99, 0.99, 0.95, and 0.85, respectively.

We then focus on a switched memory B-cell cluster composed of a group of cells labeled as benign prostatic hyperplasia and another as normal (Fig. 5A). B-cells are known to be dominant in prostate cancer and are often switched memory B-cells[83]. First, we show that they differentially express many known B-cell markers (see Supp Fig. S10). In addition, when comparing the BPH-related to the normals B-cells, we recover that the top 10 BPH-related B-cells differentially expressed genes contain many known cancer markers, B-cell markers, and a specific B-cell associated prostate cancer markers: BAG5[84] (highlighted in Fig. 5B, Table S11). Moreover, many other genes have evidence in other cancers, like CLIC4, known to be involved in the maintenance of the tumor microenvironment (TME) in breast cancer[85].

However, the number of healthy cells, especially normal memory B-cells, in this dataset is small: only 26. By performing denoising, we can recover genes that might have been missed during differential expression analysis of such a low cell count. Increasing the counts of all the genes by a factor of ten and re-doing differential expression analysis highlights some new genes whose differential expression scores are even higher than those previously cited.

Interestingly, amongst them, TSENS4, EHMT2, and IL1ORB are known to impact the function of B-cells in malignancies (see Table S11). Other genes have evidence in immunity and cancer, like TAP1, which is known to be highly expressed in immune organs and is an immuno-modulation gene known to play many roles in various cancers[86], while some genes have, of yet unknown significance, like LIP, whose paralog LIPA is a known cancer target[87] (Fig. 5B).

This demonstrates how scPRINT can embed, align, and annotate diverse datasets in a meaningful way so that one can then analyze specific and rare cell clusters to recover both known and new biology.

Finally, for the second part of the analysis, we move to another cell type of interest: fibroblasts. Fibroblasts are known to be involved in cancer[88], also called cancer-associated fibroblasts (CAFs), of which many subtypes exist, with different roles in tumor progression and invasion[89]. In our dataset, we can see a large cluster of cells labeled as "fibroblast of connective tissue of glandular part of prostate", of which 500 are coming from normal tissues, and 600 are coming from hyperplasia and are possible precursors of CAFs (Fig. 6A). Interestingly, 40% of the cells annotated as BPH-associated fibroblasts are coming from healthy tissue, according to the authors of the dataset. However, it is known that >50% of adult males over the age of 50 will have BPH[90]. Thus, one possibility is that some of the fibroblasts of these healthy tissues already present patterns of gene activation similar to those of pre-cancerous ones.

We generate a gene network of the BPH and normal fibroblasts using the 4000 most variable genes and taking the average over all heads in the network (Fig. 6A). Looking at the top 15 hubs, using degree centrality, we can see S100A6 as the top element in normal fibroblasts. This gene is known to be a fibroblast and epithelial cell marker that regulates, among other things, cell cycle and differentiation[91,92]. We also see MIF, IGFBP7, and other genes involved in immune signaling and growth[93–95].

However, some of these genes are not in common with the BPH fibroblasts ones. Over the set of 2881 common nodes between the two networks, the genes HSPA1A, MT2A, SPOCK3, ATP6V0C, DEFA1, EIF4A1, and CD99 are considered differential hubs (i.e., more central) in the BPH fibroblasts compared to normal ones (see Fig. 6A, Table S12).

Another definition of centrality, eigenvector centrality, recovers 55% of the genes already identified as hubs, plus some new ones. As an example, Prostate Associated Gene 4 (PAGE4), which is part of the GAGE family of genes, is expressed in a variety of tumors and reproductive tissues, especially BPH, where it is related to oxidative stress response and fixation (i.e., anti-invasion)[96–98]. Interestingly, although the networks share 75% of their genes, they only share 50% of their edges when considering the top 20 edges per gene. It shows that over the same set of genes, scPRINT discovers distinct gene networks across biological contexts. Taking as an example the differential hub PAGE4 (Fig. 6A), we see that it is connected to many of the top 15 hub nodes in the BPH network, such as MT2A, HSPA1A, SPOCK3, and CD99. This shows a master node sub-network linking metal and ion exchange, oxidative stress response, and inflammation[99–102]. Some genes are also part of the IL24 signaling inflammatory pathway (EIF4A1;COL6A2;HLA-C;HSPE1), and the secretory senescence phenotype (H2AZ1;UBE2S;UBE2C;IGFBP7)[93,103], hallmarks of fibrosis and malignancies[104,105]. The PAGE4 network in normal fibroblasts, while having some elements in common, like metal transport, is much less connected (seen by the strength of the edges in Fig. 6A). It also contains a different set of genes, which are less related to senescence, inflammation, and ion exchange (see Supp Fig. S11).

Furthermore, we can use these networks, defined over only a few cells, to perform community detection. Taking community 4, containing 92 genes and defined with the Louvain algorithm on the BPH-associated fibroblasts GN, we see two hub nodes: SPOCK3 and HERC3 (Fig. 6B). Interestingly, not much is known about those genes except that HERC3 has been linked to inflammation and the extracellular matrix (ECM) via metallopeptidase and the NCOA1 gene[106]. SPOCK3, moreover, is known to be related to prostate malignancies and to collagen in the ECM[107]. Gene set enrichment tells us that the genes in this subnetwork are primarily related to calcium, sodium, iron, and metal transport, validating the evidence around HERC3 and SPOCK3[108] (Fig. 6B). In normal fibroblast, however, taking the community most associated with metal transport (community 4, see details in Supp Fig. S12 and Methods) shows RNASEK, SELENOM, and an unknown ubiquitin ligase, paralog of ITCH. While RNASEK is related to RNA

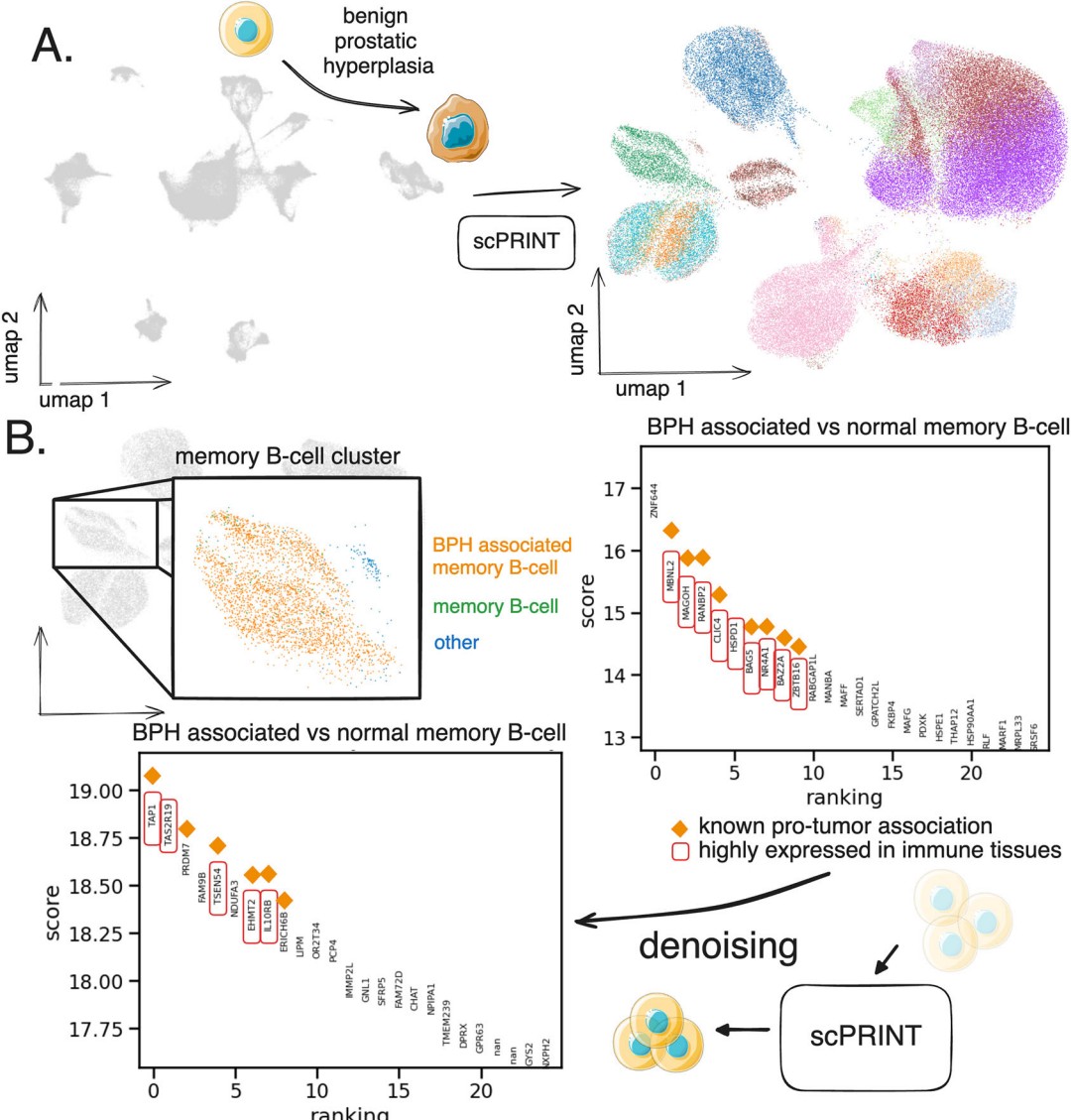

**Fig. 5 | scPRINT-based bioinformatics analysis of early prostate cancer. A** Single-cell RNAseq atlas of benign prostatic hyperplasia (BPH) and normal prostate tissues of 83,000 cells given to scPRINT. scPRINT generates a set of embeddings and label predictions for each cell. To clean our predictions, we drop cell types with <400 cells and diseases with <1000 cells, replacing them with the "other" label (see Supp Fig. S8). **B** Zooming in on one cluster, we see annotations of a switched memory B-cell cluster, some labeled "benign hyperplasia" and others "normal". Differential expression analysis on the two groups of B-cells showing enrichment of B-cell &

cancer markers when assessing its top 10 genes. We performed upsampling of the transcript count before performing a new differential expression analysis where we now see new genes amongst the top 10 differentially expressed ones some of them also associated with cancer and immune tissues. **A, B** normal-cell-2 icon by Servier https://smart.servier.com/ is licensed under CC-BY 3.0 Unported https://creativecommons.org/licenses/by/3.0/. (A): cancerous-cell-3 icon by Servier https://smart.servier.com/ is licensed under CC-BY 3.0 Unported https://creativecommons.org/licenses/by/3.0/.

degradation, its expression has been linked to a lower risk of prostate cancer[109]. *SELENOM* is of unknown function, but some SEL proteins have been related to cell adhesion[110].

Through its networks, scPRINT highlighted the role of ion exchange and fibrosis in the ECM in Benign Prostatic Hyperplasia. While some of the same genes would have been found from differential expression analysis, these results show us how gene networks can be used to describe the intersection of genes and their molecular functions. Putting genes into the context of their connections, one can validate known functions or relate them to new ones. From such contextualization, a picture starts to emerge, whereby through specific genes, glandular fibroblasts in senescence enter a wound-healing state. This fibrosis is caused by the export of more metal and ions to generate ECM and change its acidity levels. This might cause a loss in tissue flexibility and potentially create oxidative stress[111]. In our networks,

these pathways seem connected to inflammation. Chronic inflammation and wound healing states are hallmarks of BPH and a predisposition to future malignancies[112,113].

## Discussion

We can simplify the complex macromolecular interactions governing a cell through what is often referred to as a gene network. However, creating such a network in a meaningful way remains a challenging task.

We have created and benchmarked scPRINT, a novel single-cell RNA sequencing foundational model trained on >50 million single-cell profiles across tissues, diseases, and species contexts. scPRINT uses three foundational pre-training tasks, as well as new encoding and decoding mechanisms specifically designed for gene expression data. Although it has not been directly trained for it, scPRINT generates gene networks. These networks can be used to better understand the model

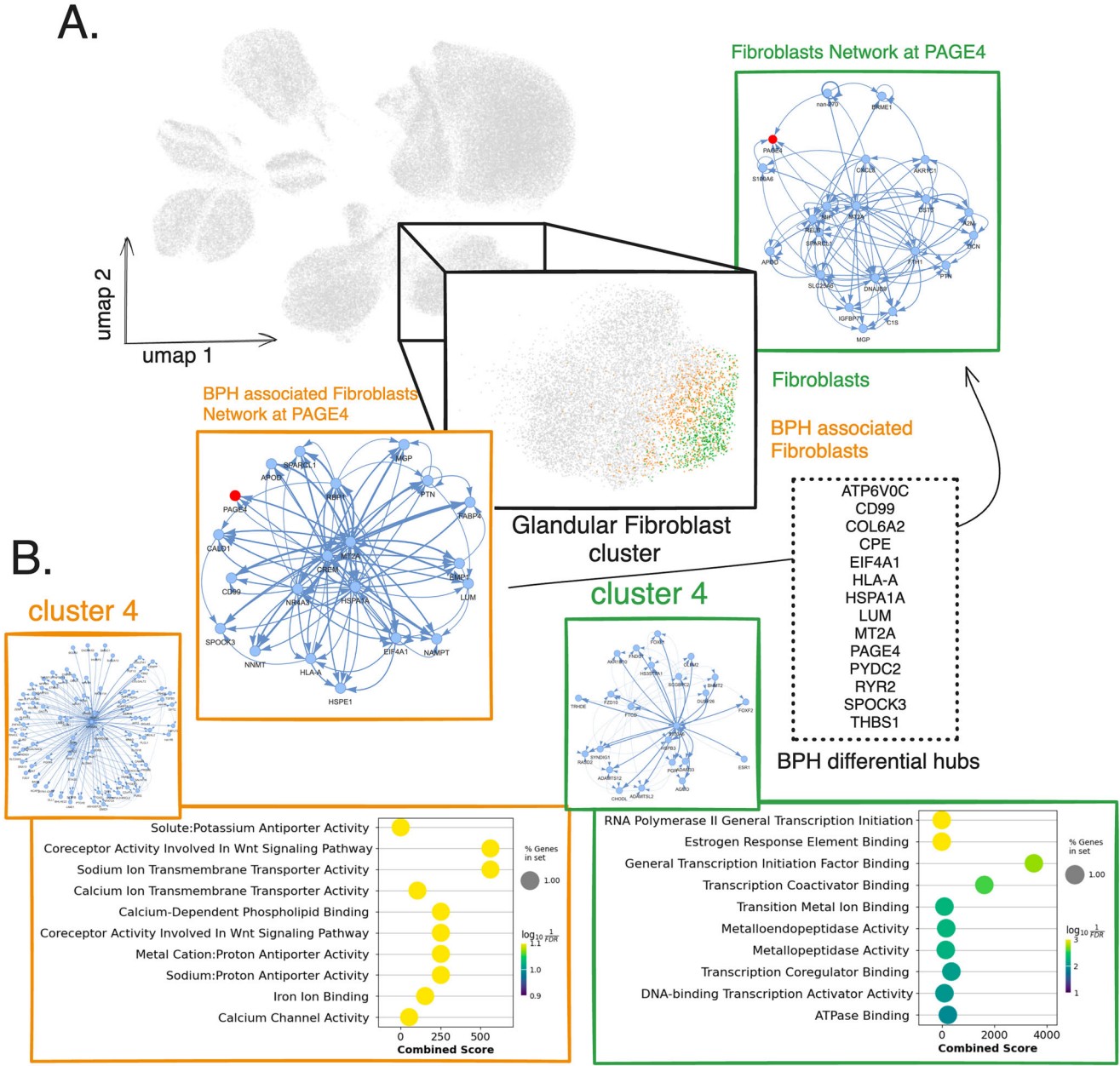

**Fig. 6 | scPRINT-based bioinformatics analysis of early prostate cancer predicts disease cell-type specific gene networks.** Continuing on the single-cell RNAseq atlas of benign prostatic hyperplasia (BPH) and normal prostate tissues of 83,000 cells given to scPRINT (**A**) Zooming in on another cluster from scPRINT's cell embeddings and annotations, we see a group labeled as "fibroblast of connective tissue of glandular part of prostate", some labeled as "benign prostatic hyperplasia", and others "normal". We generate gene networks from each and highlight a sub-network of the PAGE4 differential hub gene in BPH, showing different connection strengths and patterns between normal and BPH-associated fibroblasts. **B** Left to right: gene-set enrichment analysis, using Enrichr, of the gene community 4 found by the Louvain algorithm in the BPH-associated fibroblast gene network, same but on the normal fibroblast gene network. It shows the top 10 most strongly enriched gene sets from GO_MF_2023 according to q-value (i.e. FDR adjusted p-value).

predictions and help make more informed decisions about the significance and role of a potential target. Finally, we present a mechanism to best select heads containing the known biology of these networks. This approach also helps users fine-tune the type of network they are interested in. Given the discrepancy amongst ground truth networks, we advise users to consider using all-head averaging and to only revert to head selection when some high-confidence interactions are available. Indeed, general collections like Omnipath did not improve performance in most of our tests.

We show that we outperform other foundation models on most of our benchmarks while using a similar model size. We believe that our inductive biases and training procedures helped scPRINT achieve such

a performance. Moreover, while GENIE3 is still a competitive tool, we outperformed it on many of our benchmarks, showing that pushing training to millions of cells and large parameter sizes will be an essential direction for further work on gene network inference.

In addition, contrary to any other method assessed, our large cell model can also achieve performances on par with many famous single-cell RNAseq tools on multiple important cell biology tasks without the need for fine-tuning. While some specialized tools might be better suited to some use cases, scPRINT's versatility and speed make it a worthwhile alternative in many instances. Indeed, users can directly use scPRINT in their bioinformatics workflows with commodity hardware (1 CPU, 1 GPU with 10GB of memory and 16GB of memory).

Finally, we put scPRINT to the test on a challenging atlas of normal and senescent prostate tissues with benign prostatic hyperplasia. We identify cell populations with early markers of TME in B-cells. In fibroblasts, we study gene networks and recover known hubs such as PAGE4, thereby linking the senescence of fibroblasts to changes in the ECM and downstream inflammation. We find key interconnected pathways of the oxidative stress response and extracellular matrix building via metal and ion exchange in the gene network of BPH-associated fibroblasts. We also show that healthy and disease-related cells exhibit different network patterns, demonstrating that scPRINT can help identify novel pathways and targets while considering them in their specific cellular and molecular contexts.

An assumption in natural language processing is that fewer inductive biases make for better models. Our work shows that adding good inductive biases and rethinking architectures will likely be important directions for AI models in biology.

A challenging aspect of GN inference is that no perfect ground truths exist, and many GN methods are, unfortunately, benchmarked on ODE-generated mock-up expression data. In contrast, ChIP-seq, perturb-seq, and literature-based ground truths remain scarce and ambiguous. With BenGRN and GRnnData, our suite of tools for benchmarking Gene Networks inferred from single-cell RNA sequencing, we present an extensive set of real-world ground truths representative of the diversity of networks we can assess. However, improvement in performance and benchmarking will also need to come from innovative experimental approaches that can produce causal, genome-wide, and cell-type-specific networks containing the many different types of connections and regulations that exist, from PPI, RNA-DNA, RNA-protein, to inhibition, activation, cooperation, and more.

We acknowledge that work remains to be done, from the transformer's ability to generate graphs to their explainability and the breadth of tasks they can undertake. Questions still remain regarding the pre-training tasks and how to integrate additional data modalities into foundational models.

Transcription is much more complex than what gene networks currently represent. In the future, we expect such large cell models to work in tandem with new sequencing techniques measuring information such as time, space, protein amounts, DNA configuration, and non-coding RNA species to solve the gap in our understanding and our ability to model cell biology.

## Methods

we propose scPRINT, a foundation model designed for gene network inference. scPRINT brings novel inductive biases and pretraining strategies better suited to GN inference while answering issues in current models. scPrint outputs cell type-specific genome-wide gene networks but also generates predictions on many related tasks, such as cell annotations, batch effect correction, and denoising, without the need for fine-tuning.

### Architecture

The model architecture is composed of:
- An encoder that takes the raw data and embeds it in a high-dimensional space used by the transformer.
- A bidirectional multi-head transformer
- A decoder to transform the expression embeddings into expression values
- A decoder that transforms the cell embeddings into cell-specific label prediction over a range of classes.

**Expression encoder.** In scPRINT, each gene in a cell is converted to an embedding: It corresponds to the sum of 3 different elements:

1. An embedding representing the gene itself (see Table S2 for model embedding size). ESM2[43] embedding of each gene's most common protein product was used to represent that gene. While imperfect in some ways, this inductive bias allows the model to learn representations that potentially apply to even unseen genes from unseen species or integrate specific genetic mutations into its representation. First implemented in UCE[44], this provides the model information related to the gene product's structure, ontology, and similarity to other genes. This also speeds up the training greatly, particularly for small models. We show that this is a great gene representation, but that model performance can be increased by refining gene embeddings further during training. However, we elect not to do so to maintain the model's versatility in working on unseen genes.

We encode the genes' embeddings using ESM2. The mapping process happens the following way:
- A gene name is mapped to its canonical protein name using Ensembl[114].
- We recover the protein sequence of the protein using Ensembl
- We use the protein sequence to generate an embedding using ESM2 by averaging all the amino-acid output embeddings, as done in the ESM2 paper.

With the embedding function provided in our code, one can easily do this with any species in Ensembl.

scPRINT can effectively be retrained with any set of gene embeddings, which can be frozen during training or used only for initialization (tried, for example, in our ablation studies, Table S3).

2. An embedding of the gene location in the genome. This has also been proposed in UCE and helps the model understand that genes with similar locations tend to be regulated by similar regulatory regions[115], a relationship well-known in cellular biology.

We encode the genes' locations using positional encoding. Every gene <10,000 bp from the next is said to be in the same location; otherwise, we increment location by 1. We do this for all genes in the Ensembl database per species.

We then embed these locations by applying the Positional Encoding (PE) algorithm of Vaswani et al.[29].

3. An embedding of the gene expression in the cell. For this, we embed the gene's expression using an MLP. While GeneFormer devised a ranking strategy based on a gene expression compared to a baseline expression, scGPT instead used binning of log normalized counts. On our end, we haven't found that this approach was the simplest, nor was it performing better than only using the log-transformed counts. We thus directly take the log-transformed counts

$$\boldsymbol{e_{i,j}} = \text{MLP}(\log_2(x_{i,j}+1)), \, x_{i,j} \in \mathfrak{R}, \boldsymbol{e_{i,j}} \in \mathfrak{R}^d, \quad (1)$$

where $e_{i,j}$ is the embedding of the expression, $x_{i,j}$ is the expression value of the gene j in the cell i, and the MLP is a two-layer neural network, where each layer is composed of

$$\text{Dropout}(\text{ReLU}(\text{LayerNorm}(\text{Linear}(\boldsymbol{e_{i,j}})))), \quad (2)$$

where the Dropout rate is fixed at 0.1, and the dimensions are specified as $1 \rightarrow d$ for the first layer of the MLP and $d \rightarrow d$ for the second layer, with d representing the model dimension.

Of Note: Geneformer used positional encoding to encode gene expression, a function often used to encode the position of words in a text. Similarly to gene name token, scGPT learned an embedding for different ranges of expression values, binning them to remove sampling noise.

Both approaches apply a specific prior for the metric that defines expression. Geneformer defines expression amount as ranking based on how each gene is expressed in the cell compared to its average across all cells. Unregarding the batch effect issues, this is an assumption that expression values are not meaningful and only the

ranking of the relative abundance is meaningful information. Meanwhile, scGPT has the bias that an expression of 1, 2, or 3 are the same and that an expression 1, and 5 are different by some amount learned by the model.

By using an MLP with two layers, we effectively let the model learn the metric of transcription expression. Moreover, again, we decrease the number of parameters used compared to scGPT while being able to make predictions on count values unseen during training, such as those of bulk or pseudo-bulk RNAseq.

Finally, when encoding a cell expression profile, only a subset of 2200 genes is used during pretraining. If <2200 genes are expressed, we randomly choose 2200 expressed genes and pad them with randomly sampled unexpressed genes (meaning with an expression value of 0). This approach allows the model to see different patches of the same cell profile during training. We chose 2200 genes as 80% of the cells in cellxgene had less than this number of genes expressed, striking a balance between computation and gene usage.

We decided to add unexpressed genes because, combined with our denoising methodology, this lets the model figure out that some genes are true 0 s during training. In contrast, others are only caused by dropout and a function of the transcript counts. This causes scPRINT to model dropout as a function of read depth (i.e., total transcript count).

Moreover, this completes the minibatch by token matrix without padding and fully utilizes the GPU during the attention computation.

Of note, some models have been able to reach context lengths of 20,000 genes using the performer architecture. Performer is an often-cited method and part of the literature on attention approximation. However, most state-of-the-art transformer models do not use attention approximation as they are known to lead to worse performance[116].

Moreover, in cellxgene, more than 80% of the cells have <2200 genes being measured. This means that most of the memory and compute power is likely lost on tokens that are almost always zeros due to dropout.

The full set of embeddings of cell i sent to the transformer is the matrix $X_i$ where

$$X_i = [\boldsymbol{g_0} + \boldsymbol{e_{i,0}} + \boldsymbol{l_0}, \boldsymbol{g_1} + \boldsymbol{e_{i,1}} + \boldsymbol{l_1}, \ldots, \boldsymbol{e_{i,t}}, \boldsymbol{p_{default}}, \boldsymbol{p_{celltype}}, \boldsymbol{p_{disease}}, \ldots], \tag{3}$$

where $\boldsymbol{g_j}$ is the gene j encoding, $\boldsymbol{e_{i,j}}$ is the encoding of the expression of gene j in cell i, $\boldsymbol{l_j}$ is the gene j location encoding, and $\boldsymbol{p_A}$ is a learnt embedding for the class A.

The total count information is stored separately and encoded similarly to the expression,

$$\boldsymbol{e_{t,i}} = MLP\big(log_2(1+t_i)\big), \text{ where } t_i = \sum_j x_{i,j}, \tag{4}$$

with $x_{i,j}$ the expression value of gene j in cell i, and the MLP is a two-layer neural network similar to the previous one.

The full cell total count ($t$) lets scPRINT model its denoising based on this required total count parameter.

The placeholder tokens (total count, default cell embedding, cell type, disease, sex, ethnicity, assay, organism) are learned embeddings that stay the same across all inputs. They only act as placeholders for the model to fill in during the forward process. At the transformer's output, they will have been modified to contain the embeddings requested. At least two are used, one containing the default cell embedding and another the profile's total depth. More tokens can be used, one for each predicted cell label.

**Model.** The model is a bidirectional autoencoder similar to BERT[30] with $n$ layers, $h$ attention heads, and a dimension of $d$. It uses the flashattention2[38] methodology implemented in Triton to compute its

attention matrix. It uses the pre-normalization technique[117], with a sped-up layer norm implemented in Triton's tutorial[118]. It uses a stochastic depth with increasing dropout probability[119].

It has a 2-layer MLP with a 4 × width increase in its hidden layer and a GELU activation function.

**Expression decoder.** scPRINT uses a novel expression decoder for foundation models, which outputs the parameters of a zero-inflated negative binomial (*ZiNB*) function for each gene j in cell i. The *ZiNB* distribution is defined as

$$X \sim ZiNB(\mu, \theta, \pi), \tag{5}$$

where the parameters $\mu, \theta, \pi$ are obtained from a multi-layer perceptron (MLP) applied to the expression embeddings outputted by the transformer model at its last layer (e), which are the:

$$\mu, \theta, \pi = MLP(\boldsymbol{e}) \tag{6}$$

The MLP is a two-layer neural network with dimensions [$d, d, 3$]

Based on the work of Jiang et al.[47], zero inflation is the best distribution when considering a broad range of transcriptomic measurements, where some have enough dropouts, and a zero inflation term is needed to model it. In our case, and similarly to scVI[48], we define our *ZiNB* as

$$ZiNB(x \mid \mu, \theta, \pi) = \pi \delta_0(x) + (1 - \pi)NB(x \mid \mu, \theta), \tag{7}$$

where $\delta_0(x)$ is a point mass at zero, and $NB(x|\mu, \theta)$ is the negative binomial distribution with mean $\mu$ and dispersion $\theta$.

With these parameters, the negative binomial distribution is represented in the following way

$$NB(x \mid \mu, \theta) = \frac{\Gamma(x + \theta)}{x! \Gamma(\theta)} \left(\frac{\mu}{\mu + \theta}\right)^x \left(\frac{\theta}{\mu + \theta}\right)^\theta, \tag{8}$$

where $\mu$ is the mean and $\theta$ the overdispersion parameter, representing the inverse of the dispersion. From Hibe et al.[120], we know that this is a parameter change from the most used probability mass function (PMF) given by

$$P(X = x) = \binom{x + r - 1}{x}(1 - p)^r p^x \tag{9}$$

where r is the number of successes, $p$ is the probability of success, and $k$ is the number of failures.

One can interpret such a negative binomial distribution as a Poisson distribution with an additional overdispersion term that makes the variance not tied to the mean. In scPRINT, we use the zero-inflated Poisson for count downsampling as we can't easily infer the gene overdispersion parameter from each cell profile. By removing this zero-inflated Poisson from the gene expression profile, we keep the potential overdispersion in the profile (see the Negative Binomial to Poisson relationship section in Methods).

Compared to scVI, where the overdispersion parameter $\theta$ is learned for each gene, we make scPRINT output it together with $\mu, \pi$ (see Supp. Fig. S13).

Effectively, the model learns that the dispersion might change depending on the gene, the sequencer, the cell type, and the sequencing depth.

**Class decoder.** scPRINT also outputs a variety of class embeddings, such as default cell embedding, cell type embedding, disease embedding, etc., by filling the different placeholder tokens given as input (see the Expression encoder section in the Methods).

Effectively, for each class, we have the model learn to produce a new disentangled embedding (e.g., cell type, disease, tissue, age). This means the model uses an MLP to transform each token where A is a class. For each, we jointly train a classifier:

$$\widehat{c_A} = \sigma(\text{MLP}_A(\widehat{e_A})), \qquad (10)$$

where:

$\widehat{c_A}$ represents the logits for a class A of a dimension $d_A$ whose size corresponds to the number of labels.

$\sigma$ denotes the Sigmoid activation function.

$MLP_A$ stands for the Multi-Layer Perceptron trained to predict the logits of the class $A$.

$\widehat{e_A}$ is the output embedding for the class A of dimension $d$.

However, some classes, like cell type, have up to 800 labels. Fortunately, cellxgene classes follow an ontology, a robust structure that defines relationships among the labels. We reduce the size of the output labels by training the model only on the leaf labels in the ontology hierarchy (i.e., the most precise available). For cell types, this represents around 400 different labels (see Table S13).

Thus, when a label is not very specific for a cell type (e.g., neuron), the model will predict the best leaf label (e.g., dopaminergic neuron). This way, we can generate meaningful training signals from even very coarse labels (see The classification task section in methods for more information and definition of the loss). We only apply this hierarchical classifier to the cell type, disease, and assay labels.

In the following section, we show how we train such classifiers. During the classifiers' training, we sum up their loss without applying any scaling between the different classes.

## Ablation study

We perform an ablation study of multiple of our additions in scPRINT for its medium size version. Removing positional encoding, replacing log-normalization with a total-normalization, replacing denoising with masking, using the cell-gene product method of scGPT vs our own encoder-decoder approach to learn a cell embedding, using 2 vs 4 heads per attention blocks, not using weighted random sampling, not freezing the gene ID embeddings, and using mean-squared-error instead of the ZINB loss. For each, we re-train scPRINT entirely on the same dataset and validate its test performance with our automated benchmark platform. We provide the results in Table S3.

## Pretraining

The three tasks of the multi-task pretraining are the denoising task, the classification task, and the bottleneck learning task. While the denoising loss enhances the model's ability to find meaningful gene-gene connections, the other two try to make the model and its underlying networks more robust and cell-type-specific. All three losses are summed without rescaling.

**Optimization method**. The optimization is done with fused ADAMW, with a weight decay of 0.01. We noticed a total inability to learn when using base ADAM, which has a similar weight decay. This can be explained by a known inequivalence issue in ADAM[121].

We use the stochastic weight averaging[122] method during training with a learning rate of 0.03.

During pre-training, the hyperparameters are set to dropout of 0.1, a learning rate (LR) of 1e-4, the precision is set to 16-mixed with residuals in fp32. We clip gradients to 100 and train in many sub-epochs of 7000 training batches and 2000 validation batches with a warmup duration of 500 steps.

Across epochs, we use a linear LR decrease of 0.6 with a patience of 1 and stop training after three consecutive increases in validation loss (patience: 3). In the final layer of the class decoders, we initialize

values to a normal distribution around 1 for weights, 0 for biases, and −0.12 for biases.

Our batch size is 64, and we use a pre-norm strategy for the transformer with a linearly increasing stochastic depth dropout rate of 0.02 per layer. We use a noise parameter of 60%. We split the cells in the datasets into 98% train and 2% validation and reserve at minimum 2% of separated datasets for testing.

Finally, we use weighted random sampling on our training data based on the different class values we have to predict. We use a factor of 50, meaning the rarest elements will, on average, be sampled only 50 times less than the most common ones. The sampling factor used for each group is then $\frac{50}{count+50}$, instead of $\frac{1}{count}$ where count is the number of cells in each group.

**The classification task**. We perform label prediction during pretraining for different classes, currently: cell type, disease, sequencer, ethnicity, sex, and organism. Due to issues in the ontologies, we have omitted tissue type and age classes.

Due to the hierarchical structure of the prediction, we also created a hierarchical loss. Here, we compute the loss regularly when the label is a leaf label. Otherwise, we replace all associated leaf labels to the given label by the log-sum-exp, such that for a cell label, the loss is:

$$Loss_{classification} = CE(\sigma(\bar{c}, c), \qquad (11)$$

with:

$$\bar{c} = \begin{cases} \widehat{c} & \text{if}\{i|c_i=1\} \subseteq T \\ \text{LSE}(\widehat{c}_d)||\widehat{c}_{\sim d} & \text{else} \end{cases} \qquad (12)$$

where:

$\widehat{c}$ is the predicted vector with dimension equal to the number of leaf labels

$T$ being the set of label indices marking the labels that are leaf labels.

$\widehat{c}_d = \{\widehat{c}_i, \forall i \in T\}$ all the values in vector $\widehat{c}$ whose indices are in T. Same for $c$.

$\widehat{c}_{\sim d} = \{\widehat{c}_i, \forall i \notin T\}$ all the values in vector $\widehat{c}$ whose indices are not in T. Same for $c$.

LSE is the log-sum-exp operation

The CE (cross-entropy) is defined as:

$$CE(p, q) = - \sum_u q_u \log(p_u). \qquad (13)$$

And the LSE (log-sum-exp) is defined as

$$LSE(X) = \log\left(\sum_{p \in X} e^p\right). \qquad (14)$$

This loss allows the classifier to learn even in cases where the labels can be of varying coarseness without the coarseness of some labels impacting the ability of the model to predict the true fine-grained labels (see Supp. Fig. S14)

The loss is hierarchical for the classes: cell type, disease, sequencer, ethnicity; the labels follow a hierarchy defined by Cell Ontology, MONDO, EFO, HANCESTRO[123–126], respectively.

We do not compute the loss for cells where a class has an unknown label. We perform these classification tasks in one pass, using the embeddings generated directly from the downsampled expression profile.

**The denoising task**. Similarly to ADImpute, we expect a good gene network to help denoise an expression profile by leveraging a sparse and reliable set of known gene-gene interactions. In addition, we

expect a good cell model to help embed and reconstruct an expression profile by leveraging the regularities of modules and communities within its network.

We view denoising similarly to upsampling, and inversely, we view adding noise as downsampling a cell profile.

Noise is similar to downsampling because of the distribution we are working with. Note that contrary to vision tasks (e.g. diffusion models), where additive Gaussian noise is added, in the context of expression data, where the distribution is often seen as a Poisson, NB, or ZINB, the data is already noisy, and the more counts are sampled, the less noise. No information is similar to not sampling data.

We downsample an expression profile using a zero-inflated Poisson model of the data. With this formulation, on average, half of the counts to be dropped are dropped by randomly removing a number of reads per gene, given by sampling from a Poisson whose lambda parameter is proportional to the number of counts in that gene. The remaining half of the counts to be dropped are dropped by randomly setting some genes to 0, i.e. a complete dropout of that gene. It is to be noted that with this definition of downsampling, the exact average amount of counts dropped for both parts depends slightly on the dropout $r$. During our pretraining, $r$ is set to 0.6, meaning, on average, 60% of the transcript counts are dropped per cell.

Let $x_i$ be the gene expression vector of cell i with dimensions $n_{genes}$; we create a downsampled *version* by doing

$$\widehat{x_i} = \max((x_i - p_i) \cdot \pi_i, 0), \tag{15}$$

with:

$p_i \sim Poisson(x_i \times r \times 0.55)$ a vector of size $n_{genes}$ where the poisson is samples for each element $x_i$ of x

$\pi_i = I(u \geq r \times 0.55)$ a vector of size $n_{genes}$, the binary mask vector indicating non-dropout genes.

$u_i \sim Uniform(0,1)$, a vector of size $n_{genes}$. of random values drawn from a uniform distribution.

· denotes the element-wise multiplication.

$r$ being the dropout amount. We scale it by a tuning hyperparameter of 0.55 instead of 0.5 for numerical reasons.

The goal of the model is then using $\widehat{x_i}$ as an input to output the parameters $\mu_i$, $\theta_i$, $\pi_i$ of a *ZINB* distribution of the true profile $x_i$, all vectors of size $n_{genes}$. The contribution of cell i to the loss is then computed as the negative log-likelihood of the count data given the distribution parameters being generated by the model

$$Loss_{denoising} = Loss_{ZINB} = -\frac{1}{n_{gene}m} \sum_{i=0,j=0}^{n_{gene},m} \log(L(x_{i,j}|\mu_{i,j}, \theta_{i,j}, \pi_{i,j})), \tag{16}$$

where $n_{gene}$ is the size of the expression profile $x_i$, m is the size of the minibatch and

$$L(x|\mu,\theta,\pi) = \begin{cases} \frac{\pi}{\pi - \theta \cdot (\log(\theta) - \log(\theta + \mu))} & \text{if } x = 0 \\ \frac{(\frac{\mu}{\theta+\mu})^x \cdot \Gamma(x+\theta) \cdot \sigma(-\pi)}{\exp(\pi) \cdot (\frac{\mu}{\theta+\mu})^\theta \cdot \Gamma(\theta) \cdot \Gamma(x+1)} & \text{if } x > 0 \end{cases} \tag{17}$$

with $\sigma$ the sigmoid function.

We show that models trained with such a framework perform better than regular MSE-trained models (see Table S3), for which one only outputs one value instead of three, directly representing the data's log-transformed count. In this case, the loss is the mean squared error between the predicted and true count values.

scPRINT effectively lets the user choose between the three formulations: *ZINB* with a *ZINB* loss, NB with an NB loss, and direct log-transformed count reconstruction with an *MSE* loss.

However, we have noted that the *NB* and *ZINB* loss still have some notable issues. They can easily overflow, especially when working with lower precision systems (like fp16, bf16, etc). These losses are also proportional to the total expression count, meaning cells with higher expression will have a higher loss on average. It also appears that the log-likelihood cannot go below -1.1 loss on average and plateaus quickly. This makes evaluation of the loss less practical when comparing models. Finally, this minimal loss also depends on the total number of zeros in the true expression dataset, as the zero-inflation part of the loss converges smoothly to 0.

**The bottleneck learning task.** Bottleneck learning is a method that drives the model to generate a cell expression profile only from its embedding. Cell-embedding which can be passed again to that same model without the gene expression information, such that from the cell-embedding only, scPRINT can re-generate the cell's expression profile. The model thus finds the best compression of the cell's expression according to the information-theoretic theorem by Tishbi et al.[127].

While many transformer models and Geneformer directly use the average of gene embeddings to generate a cell embedding, this will likely squash the expression information. scGPT used another methodology (called MVC) to generate an embedding vector such that

$$x_{i,j} = e_i \odot g_j, \tag{18}$$

where $x_{i,j}$ is the expression of gene j in cell i, and $\odot$ is the dot product. For each gene embedding $g_j$, the embedding only contains information about the gene name, not gene expression. Regular MSE on each $x_{i,j}$ is then used as the training loss.

This pushes the cell embedding $e_i$ to contain all the expression information of the cell i.

This is less computationally intensive to train than our bottleneck learning method. However, we have noticed poorer reconstruction through this methodology than ours (see Table S3).

In our case, we consider that our model scPRINT can act as two parts of an autoencoder. The encoding part is when we give scPRINT the expression profile of a cell and retrieve a set of disentangled cell embeddings (see the Class decoder section of the methods). The decoder part is when we provide scPRINT only the gene labels without their corresponding expression values and the disentangled cell embedding in place of the empty placeholder embeddings (see Supp Fig. S15).

This means the encoder is considered as

$$e_{A,i} = scPRINT([g_o + e_{0,i} + l_0, g_1 + e_{1,i} + l_1, ..., p_A]), \tag{19}$$

where $e_{A,i}$ is the output embedding of the placeholder embedding token A for the cell i (in our case, we use multiple (default, totalcount, cell_type, disease, sex, organism, ethnicity, sequencer)). Then the decoder is defined as

$$\mu_i, \theta_i, \pi_i = scPRINT([g_o + l_0, g_1 + l_1, ...], e_{0,i}, e_{1,i}, ..., e_{t,i}), \tag{20}$$

With $\mu_i$, $\theta_i$, $\pi_i$ vectors of size $n_{genes}$. Finally, the loss is given by the ZINB loss:

$$Loss_{bottleneck} = \sum_{i=0}^{m} Loss_{ZINB}(x_i|\mu_i, \theta_i, \pi_i), \tag{21}$$

where $x_i$ is the cell i expression profile and $m$ the minibatch size.

Implementing a set of disentangled embeddings is not straight-forward. In our case, we push the embeddings to be as different from

one another as possible with a contrastive loss defined as

$$Loss_{constrastive} = \frac{1}{m^2} \sum_{i=1}^{m} \sum_{i'}^{m} 1 - \cos(\boldsymbol{e_i}, \boldsymbol{e_{i'}}), \qquad (22)$$

where $\boldsymbol{e_i}$ and $\boldsymbol{e_{i'}}$ are the cell embeddings, $m$ is the minibatch size, and *cos* denotes the cosine similarity. This pushes each embedding to represent the correct information using the classifiers. However, more is needed to remove all the batch effects or entirely prevent information leakage across embeddings.

Finally, we have also used the classifier output logits as cell embeddings. This works particularly well for cell type, disease, or sequencer classes containing many labels. It has been shown that classifier logit outputs behave similarly to embeddings[128] and, in our case, offer an even better removal of the batch effects (see Supp Fig. S7).

For the bottleneck loss, we directly reconstruct expression using the cell embeddings generated from the noisy, downsampled expression profile of the denoising process, doing the entire process in one single pass. We sum all the losses without scaling them:

$$Loss = Loss_{contrastive} + Loss_{bottleneck} + Loss_{denoising} + Loss_{class} \qquad (23)$$

### scDataloader

Parallel to this work, we worked with Lamin.ai to develop a dataloader for large cell atlases, described and benchmarked in Rybakov et al.[37]. One key advantage of this dataloader is its ability to perform weighted random sampling on hundreds of millions of cells without being a bottleneck during pretraining. scDataloader[129] samples cells amongst the 800+ datasets of cellxgene's mid-2023 release, using the cell labels to inform how rare the specific combination of labels is.

From this, the dataloader produces a cell sampling weight, rescaled with a hyperparameter. The dataloader will sample, with replacement, more consistently rare cell types than more common ones.

We have produced an additional wrapper package around the laminDB "mapped-dataset" called scDataloader. scDataloader works with lamin.ai but can also interface with scVI and AnnData formats to enable downloading, preprocessing, and QC of large single-cell databases and datasets. It is very flexible and can represent expression data in the formats used by scPRINT, scGPT, and Geneformer. It also implements a lightning datamodule scheme and command line interfaces for quick setup (see Supp Fig. S16).

Overall, we preprocess each of the 1200 datasets in cellxgene by only keeping primary cells from either humans or mice and dropping all the spatial omics datasets. Spatial omics are not true single-cell assays, and we decided for now not to include them. We also drop any cells with <200 expressed genes. Finally, we drop any resulting dataset <100 cells, with <10,000 genes, or from which >95% of the cells have been removed. This results in a new database of 54,084,961 cells and 548 datasets.

We believe that the weighted random sampling strategy allowed our pre-training to be much faster by creating more diverse minibatches.

### Extracting meta-cell gene networks from attention matrices in scPRINT

Transformers compute multiple attention matrices per layer, called attention heads. This is done by splitting the generated $\boldsymbol{K}, \boldsymbol{Q}$, and $\boldsymbol{V}$ embedding into $m$ sub-embeddings, thus defining $m$ attention heads. Each attention head computes the attention matrix via the equation:

$$softmax\left(\frac{\boldsymbol{Q}\boldsymbol{K}^T}{\sqrt{d_k}}\right). \qquad (24)$$

However, we would want to aggregate those over multiple cells from a similar cell state to increase the signal obtained from only one cell. We are doing so by averaging the Keys and Queries embeddings over the set of cells $U$ passed to the model:

$$softmax\left(\frac{\text{mean}_U(\boldsymbol{Q})\text{mean}_U(\boldsymbol{K})^T}{\sqrt{d_k}}\right). \qquad (25)$$

By doing this, the attention matrix behaves as if each query vector for cell i was "looking" across the key vectors of all the cells in U.

The resulting object is a row-wise normalized $n*n$ matrix, where $n$ is the size of the input context (i.e. the number of genes passed to the model). However, we also include the possibility to generate large matrices and gene networks, referred to as genome-wide gene networks. We take the average over different sets of expressed genes for each cell in the set U. This allows us to compute a genome-wide attention matrix while only doing forward passes on smaller subsets of the genome per cell.

### Heads selection

With scPRINT, we present a method to select heads based on some available ground truth data. This is inspired by the ESM2 paper[43] and uses a somewhat similar method. Using all the available attention matrices from all of the model's heads, we use a linear classifier RidgeClassifier from scikit-learn[130] (with an L2 penalty set to 1, a positivity constraint on the coefficients, and without an intercept) to classify the ground truth's edges based on a combination of each head. The classifier converts the target values into {−1, 1} equals to {no connections, connections} and then treats the problem as a regression task with mean squared error.

Instead of taking the classifier's output, we use the average of the subset of heads associated with a non-zero coefficient in the classifier, without weighting them. Thus, the classifier only serves as a means to select the heads with relevant information in predicting a ground truth of interest and decreases the possibility of overfitting (see Fig. 1C).

### Normalization and network interpretation

In scPRINT and scGPT, the attention matrix is normalized via the softmax function over the query (i.e., row) dimensions. This means that all row elements sum up to 1 or that the same mass flows from each network component. This rescaling is essential as it corrects that some row element scales can be much higher than others in the attention matrix. Similarly, in regularized models like GENIE3, only a small set of genes are connected for each gene in the matrix, meaning all genes have directed edges toward a small subset of genes. Thus, our interpretation is that the row elements are the targets in our network, each connected to a small subset of genes. The column elements are thus the regulators and can regulate many / most genes in the network.

For biological ground truths like MCalla et al. and gwps, which fit this assumption of highly connected regulators and sparsely regulated targets, we directly compare them to the inferred network. Tables S12 and S13 show that this performs better than taking the opposite view by transposing the inferred networks.

This assumption is challenged for Omnipath, which has most of its elements connected to a sparse set of other elements (see Supp Fig. S3). Due to the sparsity of connections for regulators (i.e., sources) in the ground truth network and the large number of regulators (8000+), the methods are challenged and perform much better when taking the transpose of their network and matching the regulators to the sources and sources to regulators.

### Simulated datasets, BoolODE and Sergio

BoolODE is a method to generate count data via a stochastic differential equation applied over a user-defined Boolean network. It was

used and developed as part of the BEELINE benchmark algorithm, which was created as an improvement over the GeneNetWeaver algorithm. However, this model is still very simple compared to cell biology. Due to its computational complexity, it can only model up to a couple hundred gene relationships over a few dozen genes.

Sergio[54], a slightly more recent ODE model marks an improvement over BoolODE on the size of the networks it can simulate (up to a thousand genes) and its similarity to scRNAseq data.

Indeed, Sergio's simulated data is not similar to real expression data. This means that the biases that Transformer models learn should not help them predict Sergio's data. Correlation and regression-based methods do not have biases. They are therefore expected and have traditionally shown better performance on these benchmarks.

We generated the Sergio ground truth network and simulated single cell expression by using the notebook: https://github.com/g-torr/SERGIO/blob/v2/minimal_example.ipynb from the repository: https://github.com/g-torr/SERGIO which present some debugs and improvements to the initial repository: https://github.com/PayamDiba/SERGIO. Indeed only this fork of the initial Sergio repository led us to successfully generate a network. We used RegNetwork[55] as input and simulated 1000 cells from its 3546 connections over 813 genes with default parameters from the notebook.

## BenGRN and gene network metrics

We use the packages benGRN and GRnnData released with this manuscript to work With Gene networks and perform our benchmarks.

Our three main metrics are EPR, AUPRC, and enrichment. They all take advantage of the fact that the predictions are generated as scores over edges between nodes:

– We have computed the Early Precision Ratio (EPR) as the diagnostic odds ratio: (TP x TN) / (FP x FN) at the cutoff of the scores giving $K$ positive predictions, where $K$ is the number of positive elements in the ground truth. In this context, 1 is a random prediction, and inf is a perfect prediction; values below one mean that inverting the predictor would provide better results.

– Area Under the Precision-Recall Curve (AUPRC) is the area (computed with the composite trapezoidal rule) under the curve defined by the precision ($PR = TP / (TP + FP)$) and recall ($RE = TP / (TP + FN)$) where $TP$ is the number of true positives, FP is the number of false positives, and $FN$ is the number of false negatives. This curve is obtained through a range of cutoffs going from 0 predicted positives to all predicted positives. Here, we compute a version of the AUPRC where the floor of the area is not given by the Precision = 0 line but by the line of the prevalence of the positive class. Moreover, we do not interpolate the curve between the last recall value and the perfect recall: 1. We do this to properly compare AUPRC values across benchmarks and models. Random precision values are given in the supplementary data.

– Enrichment is computed using the prerank methodology[61], where, given an ordered set of genes, it is computed by:

– Summing all scores of edges of the matrix row-wise. (Target - Hub) Or

– Summing all scores of edges of the matrix column-wise. (Regulators - Hub) Or

– Computing the eigenvector centrality of nodes in the graph[131] using NetworkX's implementation. Prerank's background comprises all the genes in the set (centrality).

Of note, we did not design an automated method for cell-type enrichment. Instead, the assessment of whether or not a network is enriched for the correct cell type is done manually, identifying cell type names in the top 10 cell types listed in the enrichment results of the network.

## Other evaluation metrics

All evaluation metrics from the section "scPRINT is competitive on tasks orthogonal to GN inference" of the results come from the openproblems benchmark and are standards in the field.

scIB's batch correction score is an average of the avgBatch score and the avgBio score, which are themselves averaged over many scores. Details of each value are available in our package's notebooks.

– scIB avgBio is a combination of label-based and label-free metrics using for example: the Adjusted Rand Index (ARI)[132] and the Normalized Mutual Information (NMI)[130] on clusters computed from the K-Nearest Neighbor graph. Other scores are used, some using the conservation of trajectories and of the cell cycle variance, and some on the rare cell population conservation, overlap of highly variable genes (see scIB[76]), and more.

– scIB avgBatch is a similar combination of label-based and label-free metrics, using, for example, the average connectivity across clusters of different batches: ASW[133], the graph integration local inverse Simpson's Index: graph iLISI[134], the k-nearest-neighbor Batch Effect Test (kBET)[133], and more.

Finally, we also use two metrics in our classification task:

– Macro-F1: also called macro-average, is the average of the F1 score across each class in a multi-class task. Where the F1 score is: $2 \times \frac{PR*RE}{PR+RE}$.

– Accuracy: the accuracy is computed as $\frac{TP+TN}{TP+TN+FN+FP}$

## Denoising benchmarks

To validate the denoising ability of scPRINT, MAGIC[72], and KNNsmoothing2[73], our test function, available in the scPRINT package, uses a representative subset of 10,000 cells of each dataset to generate the denoised expression over the 5000 most variable genes in this dataset.

Before that, counts are removed from the dataset following the same procedure as done for scPRINT's pretraining (see The denoising task section of the methods).

For each cell, we compare the denoised and un-denoised profiles to the true profile (e.g. before denoising). We compute the Spearman's correlation over the genes initially expressed in the cell, taking the average across all cells. We do not use the unexpressed genes as we are working with a dataset with high dropout and expect that a good denoiser will set genes that are 0 in the profile with some value. We notice that this improves the score of all denoising methods and makes more sense given the data.

For the rare cell population test, we keep everything similar but compute only the Spearman correlation over a rare cell population in the dataset.

We run KNNsmoothing2 with default parameters and a K of 10. We run MAGIC using the Scanpy implementation with default parameters and the approximate solver for computational speed. When computing KNNsmoothing2 or MAGIC over a small set of cells we use a K of 5 for the nearest neighbors.

## Fine-tuning

Contrary to most other foundation models for scRNAseq, we do not finetune scPRINT at any moment in our benchmark and all results are provided for the pre-trained model only.

While we haven't assessed fine-tuning we believe this is an important feature of foundation models and release various scPRINT models so that they can be re-trained, fine-tuned, and modified by the community for novel tasks or to improve its performance on the tasks we have presented.

## State-of-the-art methods used in benchmarking

All methods presented here generate networks from their input data. Given gene-level expression data, they will generate gene-networks.

Without additional information, no method can distinguish the type of molecular interactions that underpin their predicted network edges.

**Gene network inference with an ensemble of trees (GENIE3).** Developed originally for bulk transcriptional data, *GENIE3* computes the regulatory network for each gene independently. It uses a random forest, a weak learner ensemble method, to predict the expression profile of each target gene from profiles of all the other genes. The weight of an interaction comes from the feature importance value of an input gene in the predictor for a target gene's expression pattern. Aggregating these weighted interactions over all the genes yields the regulatory network. This method was the top performer in the DREAM4 in silico network challenge (multifactorial subchallenge).

*GENIE3* can be seen as a generalization of correlation-based methods for inferring gene networks. Instead of looking at genes that correlate most with another gene, GENIE3 finds how to combine a set of correlated genes to get an even better correlation. We run GENIE3 on raw counts as it is said from both the BEELINE benchmark and the R package vignette that GENIE3 can be run on either log normalized or raw count data and that while it will change the results, there are no preferred methods. This is something we have also noted in our trials. We use all default parameters and choose 100 trees for computational feasibility reasons. We compute the networks on the same set of cells and genes as the other methods.

We also use a TF-gene only version of the method where the regression is performed only using the expressed transcription factors instead of all expressed genes as input. This is the most used version of *GENIE3* and is much faster.

**DeepSEM.** DeepSEM is an autoencoder model made for gene network inference. It learns to decompose a set of cells as a set of embedding and an adjacency matrix (i.e., a gene network). The formula of the VAE then becomes: $X = f_1((I - W^\top)^{-1}Z)$, for the decoder and $Z = (I - W^\top)^{-1}f_2(X)$ for the encoder, where $X$ is the expression data, $Z$ is the embedding dimension, $W$ is the adjacency matrix, $I$ the identity, and $f_1, f_2$ are MLPs.

We preprocess the anndata by normalizing gene expression to 10,000 genes, applying a logp1 transformation, and then computing the z-score per gene, as explained in the associated research paper. We use DeepSEM with default parameters and on the same set of cells and genes as the other methods. We use the DeepSEM-provided functions for loading and parsing Anndatas.

**Single-cell generative pretraining transformer (scGPT).** scGPT is a transformer-based model of roughly 100 M parameters, pre-trained with a generative process similar to Language models. scGPT proposes to build similarity networks based on the output gene embeddings of the model but also based on its attention matrices. It computes networks as the difference between the rank-normalized version of the average attention matrix in a baseline expression profile vs a perturbed one in perturb-seq data. The attention matrix is the average of attention matrices over the heads of the last layer and over the cells given to the model.

We run scGPT following the examples given in their "Tutorial_Attention_GRN.ipynb" notebook.

We use the "scGPT_human/best_model.pt" from the list of available models with default parameters. All runs are in our fork: "https://github.com/jkobject/scGPT" in the "mytests/" folder. Similarly, we take the mean over cells and over the heads of the last layer. We compute softmax similarly to the attention computation but without applying the rescaling factor $\sqrt{d_k}$. We finally drop the first element corresponding to the cell embedding token.

We extract cell embeddings from scGPT by directly using the cell embedding token of the model without fine-tuning it on a batch correction task. This is done in order to compare it to scPRINT which is itself not fine-tuned. We compute the networks on the same set of cells and genes as the other methods.

**Geneformer.** Geneformer is a BERT model. Gene expression data is transformed into a sentence or genes ordered by their scaled expression. It is trained with mask language modeling and contains somewhere around 80 M parameters. We use the new versions of 2024 Geneformer models[52] trained on 100 M cells (2 × more than scPRINT). We follow the preprocessing and inference scripts used in the geneformer huggingface repository and notebooks: https://huggingface.co/ctheodoris/Geneformer/tree/main. Our inference script updates to extract gene networks from Geneformer are available in our scPRINT repository: https://github.com/cantinilab/scPRINT/tree/dev/tools.

We extract gene networks from Geneformer using the mean of all attention heads per cell. Since Geneformer only uses expressed genes in a cell, we have to map the attention matrices back to the full network size before computing its average over cells, taking into account the NaN values. We compute the networks on the same set of cells and genes as the other methods.

We extract a cell embedding from Geneformer using the cell embedding from the "gf-12L-95M-i4096_MTLCellClassifier_CELLxGENE_240522" model that has been fine-tuned on predicting the cell labels of cellxgene datasets.

**scFoundation.** scFoundation is a foundation model for single-cell RNAseq based on the xtrimogene architecture[135]. It was built by the Biomap company. It is able to work on the full genome sequence of transcripts for each cell by considering the high number of zeros and embedding them separately. The tool is aimed at performing a range of tasks, such as denoising, embedding, and predicting perturbation response. It has been trained with a mixed masking and denoising pre-training. However, we could not compare scFoundation to scPRINT and MAGIC on the denoising benchmark, as scFoundation's denoising only happens at the level of the cell embedding at inference time.

We could not validate scFoundation on our Gene network inference benchmark as extracting a network from the attention matrices was much more complex due to the xtrimogene architecture. scFoundation mentions the generation of gene modules using clustering of its output gene embeddings. It also mentions the interference of gene networks. However, it is achieved using RcisTarget[8], a prior gene network based on motif analysis. This approach is not comparable to the gene networks generated by scPrint, Geneformer, and scGPT. Indeed, RcisTarget could be applied to every model we have benchmarked and would prevent us from doing an unbiased benchmark. Neither our approach nor Hao et al.'s could extract gene networks directly from scFoundation. It is being left to further investigations.

For batch effect correction, we use scFoundation with default parameters and follow the steps for cell embedding in the "model/README.md" file in their GitHub repository: https://github.com/biomap-research/scFoundation. However, we give scFoundation single cell profiles of the 5000 most variable genes in each dataset. This is because we could not run scFoundation on genome-wide expression profiles with our GPU. We then apply a PCA to the output embedding to reduce the dimensionality from 3224 to 512. This is because the initial dimension was too high for scIB to compute a score from on our machine (40CPU Intel Xeon, 32GB RAM + 64GB SWAP, GPU NVIDIA A4500 with 20GB of memory).

**Marker-based cell type prediction with CellTypist.** To showcase the novel ability of scPRINT to perform zero-shot prediction of cell type labels, we use the CellTypist method, which similarly performs de-novo prediction of cell type labels given its precomputed databases of cell type markers.

CellTypist works by mapping cell gene expression to genes known to be specifically expressed in combination in a cell type. Thus, it predicts cell type from these marker genes.

We use it with default parameters on the normalized and log-transformed counts over the full set of genes in the dataset. We use the "Adult_Human_PancreaticIslet" database, which contains markers for 14 cell types and overlaps with only four of the cell types in the dataset.

We decided to still use it as is to showcase the marker-based method's inability to recover the full set of cells and the tradeoff between the number of cell types and accuracy.

Fortunately, these four cell types (A, B, D, PP) represent 70% of the dataset. With its current database, CellTypist can only reach a maximum accuracy of 70%. Even when taking this into account, CellTypist only overperforms scPRINT on the accuracy metric and by roughly 9 points.

**Classification benchmark and associated methods.** Our classification benchmark is run using following the openproblems benchmark. It uses the same input, output data, and metric. It also similarly splits the train-test by batch and preprocesses the expression matrix to what is presented in the open problem benchmarks.

For this task, methods can access the full set of genes by default. scPRINT will use its random sampling of genes approach with a context of 4000 genes. Classifiers like logistic regression and xgboost were run according to the openproblem process, using the 25 principal components of the count normalized, logp1 transformed expression data. CellTypist was run on the normalized and logp1-transformed cell expression profile.

### Ground truth preparation

**McCalla et al.** For the MCalla et al. dataset, we downloaded the data from the supplementary datasets of their paper https://www.biorxiv.org/content/10.1101/2021.06.01.446671v2.supplementary-material. After undoing the logp1 transform, we re-generate the true count expression matrix from the normalized one by dividing the expression of each cell by the smallest value in its expression profile. This fully recovered the true counts, all values being integers. For the additional human dataset we used, we downloaded it from the gene expression atlas database https://www.ebi.ac.uk/gxa/sc/experiments/E-GEOD-36552/downloads.

We used the intersection (gold standard) ground truth dataset for both human and mouse, converting this list of sources to target genes into a directed binary network.

**Omnipath.** We generate the Omnipath network using all the interactions from the Omnipath Python package, excluding small molecules, lncRNAs, and any element without a unique HGNC symbol. We then transform it into a directed binary network of source to target. These interactions are extracted from the literature and represent mainly TF to gene connections as well as many protein-protein interaction connections and a small number of other connections known from the literature like RNA-RNA interactions, protein-RNA interactions, and more. All interactions are mapped back to their gene IDs, generating a gene-gene network encompassing the various interactions the genes and their molecular products can have.

**Gene networks from genome-wide perturb-seq.** We created a gene network from the genome-wide perturb-seq dataset using the supplementary matrix containing the results of differential expression in the dataset. This matrix represents the multiple hypothesis testing corrected $p$-values of a differential expression test of cells with KO of gene A compared to the baseline cell expression. This is available for all 8000+ expressed genes in the K562 cell line. We used a cutoff of 0.05 on these values to define the directed binary connection between genes.

This effectively gives a gene x gene-directed binary graph that tells if a statistically significant connection exists from the source $gene_A$ to the target $gene_B$ according to genome-wide perturb-seq.

For all ground truths, download, preprocessing, and extraction of the network and expression data are available in the BenGRN package.

### Details on the benign prostatic hyperplasia analysis

We download our dataset from cellxgene under the reference: 574e9f9e-f8b4-41ef-bf19-89a9964fd9c7.

We preprocess the dataset using scDataloader's preprocessing function. We generate embedding and classification using 3000 expressed genes in each cell. Similarly to pretraining, we take 3000 randomly expressed genes; if <3000 are expressed, we complete with randomly selected unexpressed genes. We display embeddings generated using the cell type classifier logits (see section The classification task in methods)

We use the Scanpy toolkit[136] to generate our Umap plots directly from the embeddings, as well as our differential expression results and our clusters. We define the clusters using the Louvain algorithm with 10 k-nearest-neighbors and a resolution of 1. We perform denoising on 5000 genes per cell selected similarly to the embedding and classification part. We use the 4000 most variable genes in each cell type to generate our gene networks in the BPH and normal fibroblasts.

On the gene networks, we perform gene set enrichment with the Enrichr method[137] on the GO_MF_2023 gene sets. For community detection, we use the Louvain algorithm with parameter 1.5. We perform analysis only on the communities with between 200 and 20 genes. (4 and 5 in the BPH-associated fibroblasts, 3 and 4 in the normal fibroblasts)

All analysis and results are available in the *cancer_usecase_1* and *cancer_usecase_2* notebooks.

### Negative binomial to Poisson relationship

As explained in The denoising task and Expression decoder section of the methods, in our model, we have used the ZINB as our loss, an extension of the NB distribution to zero-inflated data. Moreover, we have also used the zero-inflated Poisson mechanism to downsample the cell expression profiles. These are consistent because we can view the Poisson distribution as a NB without overdispersion. The relationship between *NB* and *Poisson* is given by making the dispersion term go to 0 and the inverse dispersion term $\theta \to \infty$. Doing so, the term $\frac{\theta}{\theta+\mu}$ approaches 1. Thus, the PMF simplifies to:

$$P(X=x) \approx \frac{\Gamma(x+\theta)}{x!\Gamma(\theta)} 1^\theta \left(\frac{\mu}{\theta+\mu}\right)^x \tag{26}$$

For large $\theta$, we use Stirling's approximation[138,139] of the Gamma function: $\Gamma(\theta) \approx \sqrt{2\pi\theta}(\frac{\theta}{e})^\theta$ we get:

$$\Gamma(x+\theta) \approx \sqrt{2\pi(x+\theta)}\left(\frac{x+\theta}{e}\right)^{x+\theta} \tag{27}$$

$$\Gamma(\theta) \approx \sqrt{2\pi\theta}\left(\frac{\theta}{e}\right)^\theta \tag{28}$$

Simplifying the ratio of the Gamma functions:

$$\frac{\sqrt{2\pi(x+\theta)}(\frac{x+\theta}{e})^{x+\theta}}{\sqrt{2\pi\theta}(\frac{\theta}{e})^\theta} = \sqrt{\frac{x+\theta}{\theta}}\left(\frac{x+\theta}{\theta}\right)^\theta\left(\frac{x+\theta}{e}\right)^x. \tag{29}$$

For large $\theta$, $\frac{x+\theta}{\theta} \sim 1$, so:

$$\sqrt{\frac{x+\theta}{\theta}} \approx 1$$

$$\left(\frac{x+\theta}{\theta}\right)^{\theta} \approx 1$$

Thus, the expression simplifies to:

$$P(X=x) \approx \frac{1}{x!}\left(\frac{\mu}{\theta+\mu}\right)^x \left(\frac{\theta+x}{\theta}\right)^x \quad (30)$$

Finally, $\left(\frac{x+\theta}{\theta+\mu}\right)^x \approx 1$ for large $\theta$, so:

$$\lim_{\theta \to \infty} P(X=x) = \frac{\mu^x}{x!}e^{-\mu} \quad (31)$$

This is the PMF of the Poisson distribution with mean $\mu$.

## Reporting summary

Further information on research design is available in the Nature Portfolio Reporting Summary linked to this article.

## Data availability

The model weights are publicly available on huggingface under: https://huggingface.co/jkobject. Pre-training logs to assess the model's training are publicly available in weights and biases under: https://wandb.ai/ml4ig/scprint_scale/reports/scPRINT-trainings--Vmlldzo4 ODIxMjgx?accessToken=80metwx7b08hhourotpskdyaxiflq700xzm zymr6scvkp69agybt79l341tv68hp. The full pre-training dataset is publicly available on CellxGene: https://cellxgene.cziscience.com/ under its census data release version: LTS 2023-12-15. All of the other datasets used in this work can be downloaded through their respective public databases via the helper scripts on the scPRINT, BenGRN, GRnnData, and scDataLoader packages. Source data are provided with this paper to re-generate the figures. Code to generate the large Umap of Fig. 1 is available as a notebook in Github: https://github.com/cantinilab/scPRINT/blob/1.6.4/figures/nice_umap.ipynb. Code to re-generate the Source data is available as notebooks in our Github: https://github.com/cantinilab/scPRINT/tree/1.6.4/notebooks and listed in our inventory of supplementary information. Source data are provided with this paper.

## Code availability

The code and notebooks used to develop the model, perform the analyses, and generate results in this study are publicly available and have been deposited in cantinilab/scPRINT at https://github.com/cantinilab/scPRINT under MIT license. The specific version of the code associated with this publication is archived in the same repository under the tag 1.6.4 and is accessible via https://github.com/cantinilab/scPRINT/tree/1.6.4/ and https://doi.org/10.5281/zenodo.14749466[33]. Additional developed packages for this analysis are defined in the pyproject file and project submodules. They are available on GitHub: • GRnnData: https://github.com/cantinilab/GRnnData https://doi.org/10.5281/zenodo.10573141. • BenGRN: https://github.com/jkobject/benGRN https://doi.org/10.5281/zenodo.10573209. • scDataLoader: https://github.com/jkobject/scDataLoader https://doi.org/10.5281/zenodo.10573143. • scGPT and notebooks to reproduce the results: https://github.com/jkobject/scGPT/tree/main/mytests.

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

## Acknowledgements

The project leading to this manuscript has received funding from the Inception program (Investissement d'Avenir grant ANR-16-CONV-0005) L.C. and the European Union (ERC StG, MULTIview-CELL, 101115618) L.C. We acknowledge the help of the HPC Core Facility of the Institut Pasteur and Déborah Philipps for the administrative support. L.C. The work of G. Peyré was supported by the French government under management of Agence Nationale de la Recherche as part of the 'Investissements d'avenir' program, reference ANR19-P3IA-0001 (PRAIRIE 3IA Institute) G.P.

## Author contributions

J.K., L.C., and G.P. designed the study. J.K. developed the tool, and performed all the analysis. J.K., and L.C wrote the manuscript. G.P. and J.S. revised the manuscript. J.S. gave feedbacks during the entire project and reviewed the code and its usability.

## Competing interests

The authors declare no competing interests.
