## [Transparent Peer Review file · Nature Communications]

scPRINT: pre-training on 50 million cells allows robust gene network predictions

Corresponding Author: Dr Laura Cantini

Version 0:

Reviewer comments:

Reviewer #1

(Remarks to the Author)

The authors build a pre-train model using 50M cells' transcriptome data. The authors aim to show that the proposed large pre-trained model can better predict gene networks. My comments are shown as follows:

1. It is unclear which kind of interactions in a gene network scPRINT can infer. The primary goal of the paper is to show that scPRINT outperforms other methods on gene network inference. However, the definition of a gene network is unclear. As mentioned in lines 26-31, different kinds of interactions exist in a gene network (TF-Gene, RNA-RNA, and protein-TF). It is unclear whether scPRINT can infer all kinds of interactions in a gene network or just TF-Gene interactions or other kinds of interactions. In experimental section 1 (line 154), the authors showed scPRINT can infer all kinds of interactions (such as TF-gene, protein-protein interaction, etc) provided in Omnipath. However, in experimental section 2 (line 249), the authors showed scPRINT could infer TF-gene regulation (the ground truth is provided by pertrub-seq and ChiP-seq). The authors need to clarify the exact interactions scPRINT can infer.
2. Following up on the first point, in experimental sections 1 (line 154) and section 2 (line 249), scPRINT takes scRNA-seq as input but outputs different kinds of interactions. In section 1, scPRINT infers all kinds of interactions. In section 2, scPRINT infers TF-gene regulations. Is there a way to know which kind of interaction scPRINT would output based on the input?
3. There are other foundation models built based on large single-cell data, such as scFoundation and biolord. The authors should include them in all the experiments.

(Remarks on code availability)

Reviewer #2

(Remarks to the Author)

Cell-specific gene network prediction can provide insights into cellular functions and behaviors from a systems perspective. In this study, the authors proposed a pre-training model, called scPRINT, which is a foundation model designed for the inference of gene network. The primary distinction of scPRINT lies in representing gene expression in a specific cell using ESM2 embeddings, incorporating gene expression embeddings and gene position embeddings. This approach utilizes a bidirectional multi-head transformer to construct and identify associations between genes. Numerous experiments and tests were conducted to validate the accuracy and biological significance of the inferred network. It is an interesting exploration in the area of foundational gene network models. The methods section provides a comprehensive overview of the experimental design and analytical procedures used in the study. The study also provides the code for the methods and some replication code. However, the manuscript has several limitations that need further explanation and clarification:

1) When validating on the Omnipath network, the authors mention that scPRINT-omni "carefully split the dataset into train/test and select, using 50% of the ground truth on the first cell type of each dataset," which suggests a supervised approach to selecting attention heads. However, the authors do not describe how GENIE3 and scGPT handle this. Based on my understanding of GENIE3 and scGPT, they likely use an unsupervised approach to construct gene networks. This could

result in an unfair comparison.

2) The manuscript defines the gold standard for gene networks differently, such as the Omnipath literature-based ground truth and Mcalla et al (ref 24). Many metrics, like AUPR and EPR, heavily depend on the accuracy of the gold standard. When the gold standard is inaccurate, these metric comparisons become meaningless. The authors might consider incorporating reliable simulated datasets, such as BoolODE from BEELINE, for unbiased comparison.

3) The sentence "the Omnipath literature-based ground truth does not reflect the topology of a biological network" is unclear. Why doesn't it reflect network topology?

4) The primary contribution of this paper is the construction of cell-specific gene networks. Previous studies have referred to these cell-specific gene networks as dynamic networks (see Citation1 and 2), often utilizing them in dynamic processes such as cell development and differentiation. However, this paper does not address the analysis of changes in gene networks during processes like cell development and differentiation, which I believe is a significant shortcoming.

[1] Nature Communications, 2023, 14(1): 8459.

[2] Nature Methods, 2023, 20(9): 1368-1378.

5) Large foundation models are not the only solution for constructing gene networks based on single-cell RNA-seq. Recently, there have been numerous methods specifically designed for gene network construction using scRNA-seq that do not rely on pre-training[1],[2],[3]. The authors need to further demonstrate that scPRINT, which utilizes large model pre-training, is superior to these methods tailored to specific datasets. Comparing with GENIE3 alone is insufficient, as GENIE3 is a method from over a decade ago and was not specifically developed for scRNA-seq data.

[1] Nature Computational Science, 2021, 1(7): 491-501.

[2] Nucleic Acids Research, 2023, 51(4): e20-e20.

[3] Briefings in Bioinformatics, 2023, 24(5): bbad326.

6) "In contrast, head selection is often more advantageous in larger models with more heads (see Table S4)", Table S4 is missing in the supplementary file.

7) In line 342, the authors mention, "While scPRINT does not train on the test dataset itself and makes predictions over hundreds of labels, it still reaches 62% classification accuracy (Figure 4B, Supp Figure S5)." However, the comparison methods chosen were logistic regression and xgboost. Why was scGPT not selected for comparison? 62% classification accuracy is not particularly high, even without fine-tuning.

8) According to the methods section, the embedding for a specific gene expression includes ESM2 embeddings, gene expression embeddings, and gene position embeddings. However, the ESM2 embeddings and gene position embeddings are identical for the same gene. The variation in attention calculation across different cells mainly depends on the gene expression embeddings. Since the embedding for expression is merely an MLP-upscaled version of the log-transformed expression value, I have some doubts about the effectiveness of this representation approach.

9) "Taking as an example the differential hub PAGE4 (see Figure 4C), we see that it is connected to many of the top 15 hub nodes in the BPH network," This should be Figure 5.

10) In gene network prediction, the characteristic of gene relationships, such as activation or inhibition, is also crucial. The authors should discuss whether scPRINT can potentially infer the characteristic of these gene network relationships.

(Remarks on code availability)

The GitHub repository provides installation instructions for scPRINT and key reproducibility code, which can be successfully installed. It also includes detailed API documentation.

Reviewer #3

(Remarks to the Author)

Summary

The authors propose scPrint, a foundation model that is pre-trained on 50 million single-cell profiles, with a focus on gene network inference and downstream tasks such as cell type prediction, batch effect correction and denoising. In particular, the use of inductive biases via a denoising-based pre-training task improves downstream model performance across multiple benchmarks discussed in the manuscript. Their work further advances the field of developing foundation models in bioinformatics and provides improved performance across various benchmarks in GRN inference.

Major comments

1. It would be helpful to have a guideline on the general recommendations as to which type of scPRINT model to use in practice. The figures in 2B suggest the use of scPRINT-mean or scPRINT-omni, while I believe in 3B, a different version of scPRINT(-self) is more appropriate. In particular, in 3B, it seems that only scPRINT-self can "outperform" random guessing. The performance appears better in Figure 3C, but we see a more varied result there, with Genie 3 being competitive. How do the scPRINT models used in Figures 2/3 compare to the scPRINT models in Fig. 4?

2. Gene embedding that aggregates multiple lines of evidence is interesting. However, it's unclear how different types of embedding results play a role in downstream analysis. Especially how AA sequence information and genomic positional embedding modulate gene expression embedding patterns are worth investigating.

3. The selection of attention heads for scPRINT could be explained more clearly, together with its implications on interpretability and further downstream usage.

4. What type of pretraining tasks were performed? Sure, we can speculate based on previous work (and figures), but it is not fully stated in the texts. Were these different tasks combined or separate and sequential?

Minor comments

1. Please consider using different fonts in the main figures. Some parts are not clearly legible.

2. Why 2,200 genes rather than 20k genes? Other methods take advantage of full coding genes (e.g., Hao .. Song, Nature Methods, 2024). I think embedding could handle a flexible number of genes even though many of them were cropped in cell x gene DB.

3. Comparison with other ML models needs more clarity. What the training and testing data sets were and how hyperparameters were tuned could certainly change the performance.

4. Non-transformer models are expected to perform worse, but it would be worth confirming that these methods also used full-scale data (with random sampling).

(Remarks on code availability)

Reviewer #4

(Remarks to the Author)

(Remarks on code availability)

Version 1:

Reviewer comments:

Reviewer #1

(Remarks to the Author)

I first want to thank the authors for responding to my question in detail. But I still have some concerns:

1. Regarding the interaction type that scPRINT can predict. Can you give some examples of how protein-protein interactions and protein-TF interactions can be predicted if we only input gene expression to the scPRINT? One gene generates multiple proteins. It is challenging to predict those interactions using only gene expressions.

2. In scFoundation paper, it shows that it can predict TF-gene interaction in the last section. You should compare scPRINT with it.

(Remarks on code availability)

Reviewer #2

(Remarks to the Author)

The author have addressed all my questions in the revised version

(Remarks on code availability)

Reviewer #3

(Remarks to the Author)

I believe the authors addressed all the comments that we have raised in the previous round. We do not have any further

comments.

Congratulations on your interesting work!

(Remarks on code availability)

Reviewer #4

(Remarks to the Author)

(Remarks on code availability)

Point-by-Point Reply to reviewers

Reviewer #1:

1. It is unclear which kind of interactions in a gene network scPRINT can infer. The primary goal of the paper is to show that scPRINT outperforms other methods on gene network inference. However, the definition of a gene network is unclear. As mentioned in lines 26-31, different kinds of interactions exist in a gene network (TF-Gene, RNA-RNA, and protein-TF). It is unclear whether scPRINT can infer all kinds of interactions in a gene network or just TF-Gene interactions or other kinds of interactions. In experimental section 1 (line 154), the authors showed scPRINT can infer all kinds of interactions (such as TF-gene, protein-protein interaction, etc) provided in Omnipath. However, in experimental section 2 (line 249), the authors showed scPRINT could infer TF-gene regulation (the ground truth is provided by pertrub-seq and ChIP-seq). The authors need to clarify the exact interactions scPRINT can infer.

Reply: We realize from the reviewer's comment that our description of scPRINT's network inference was not evident in the current version of the paper. scPRINT generates networks containing any type of interaction, regardless of its input.

At the same time, scPRINT can produce two types of networks:

- One uses the mean of all attention heads (now simply called *scPRINT*), which is not specific to any interaction type (we call it a gene-gene network).
- One uses the mean of selected attention heads using some ground truth information (now referred to as *scPRINT([ground truth name]'s heads)*, e.g., *scPRINT(omnipath's heads)*, when using *omnipath* for selecting the heads. This version might be more specific to the interaction types present in the ground truth.

Both can be subsetted to TF-gene networks (GRNs) by removing non-TF ->gene links. This is the same approach we have now used for DeepSEM and Geneformer.

For GENIE3, however, we have to run it twice. Once using all genes as predictors and once using only TFs.

As a reminder, we select attention heads as presented in the ESM2 paper, using the selected features of a linear classifier over the attention heads.

This selection aims to specialize the network to specific interaction types (e.g., literature ground truth information or perturbation-based networks). Unfortunately, selecting heads requires some amount of ground truth information that is missing or too scarce to specialize the network to specific interaction types.

Finally, the networks are often generated over a set of selected genes, e.g., the 5000 most variable genes. However, they can also be made over the entire genome, called the

scPRINT(genome), in section 2 of the result.

- We have changed the end of the first section of the results.
“The attention heads are either all aggregated by averaging or can be selected to better reflect connections of interest (Figure 1C). This is done using the average of the heads most correlated with literature or perturbation-based ground truth networks. Finally, while we do not assess scPRINT’s ability to model inhibition due to the scarcity of such annotations, we leave open the possibility of using our head selection technique for such a task.”
- We improved our explanation of the output of scPRINT across the different tests: “Of note, due to the small number of genes assessed in the ground truth, we do not add the genome-wide network versions here. ”, and “Moreover, only the TF version of GENIE3 and the TF-gene subsets of the other method’s networks are used since the ground truth only contains TF-gene connections.” in section 3 of the results, and “For scPRINT, we generate three network versions: one simply called *scPRINT*, based on the average of all heads in the model. *scPRINT (omnipath’s heads)*, based on the average of heads selected with our abovementioned head selection method inspired by ESM2, and *scPRINT (genome)*, which is like the *scPRINT* network but uses our method to generate genome-wide networks (see Methods) instead of using the 5000 most differentially expressed genes.” in section 2 of the results.
- In the 7th paragraph of the discussion, we discuss the limitations of scPRINT in distinguishing the types of interactions it infers (due to the scarcity of ground truth).
- We have renamed scPRINT-self to scPRINT(Han et al.’s head): heads selected with the Mc Calla et al. ground truth, scPRINT(gwps’ heads): heads selected with the gwps ground truth, and scPRINT(omnipath’s heads): heads selected with the omnipath ground truth and updated the figure legends.
- we make it clear that both are the same for gene networks and gene regulatory networks in paragraph 9 of section 2 of the results

2. Following up on the first point, in experimental sections 1 (line 154) and section 2 (line 249), scPRINT takes scRNA-seq as input but outputs different kinds of interactions. In section 1, scPRINT infers all kinds of interactions. In section 2, scPRINT infers TF-gene regulations. Is there a way to know which kind of interaction scPRINT would output based on the input?

Reply: We understand that there is a misunderstanding about this part of the analysis. From the answer to question 1, we hope to have clarified that scPRINT can generate any network type regardless of its input.

However, we selected networks based on the task.

In Figure 2, Omnipath contains all types of connections and a TF-gene-specific subset. We have thus used both a gene-gene network and a TF-gene network to compare it to. While for scGPT and scPRINT, this comparison only involves subsetting the network to TF-gene elements, for GENIE3, it involves entirely recomputing the network, letting it only select TFs, which will completely change the network.

For Figure 3B In McCalla et al., the connections are only TF-genes, meaning we already only assess the TF-gene connection subset of scPRINT and scGPT. We removed the GENIE3 version on all genes, as only the gene-TF makes sense. We hope it will avoid confusion.

For Figure 3C, genome-wide perturb-seq, most genes, including TF-coding ones, were knocked out. For this reason, gene-gene networks were compared. Based on the reviewer's comment and to better separate GNs and GRNs, we added a supplementary figure in which we only test on the TF knock-out subset of the network and display only the gene-gene network assessment in Figure 3C.

- We have added a supplementary figure (Figure S4) showing results on the TF-gene-only version of the GWPS network.
- We have changed paragraphs 3 and 5 of the third section of the results. "Of note, due to the small amount of genes assessed in the ground truth, we do not add the genome-wide network version here. Moreover, only the TF version of GENIE3 and the TF-gene subsets of the other method's networks are used since the ground truth only contains TF-gene connections.", and "Based on both AUPRC and EPR, scPRINT outperforms all other methods on this benchmark (Figure 3B). This means, for example, that when training GENIE3 to only predict a gene's expression based on TF expressions, it is not selecting the right TFs amongst the set of a few dozen assessed in McCalla et al..

scGPT, Geneformer v2—and, in a few cases, scPRINT—can have values worse than random guessing. Thus, their predictions are often specific to some TFs but not necessarily the right ones (Figure 3B).

- In paragraph 10 of section 2 of the results, we explain more clearly that the TF-gene-only versions of the scPRINT and scGPT networks are only a subset of their gene-gene networks. "However, GENIE3 is often used by biasing the method to only predict TF-gene connections (see Methods). This type of network, usually called a gene regulatory network (GRN), is most often used, given the importance of TFs in regulating gene expression. To compare the other methods to the GRN version of GENIE3, we also use a "GRN" version of their networks by subsetting them to TF-

gene connections only. In this context, all the methods significantly improve their predictions without significantly altering their relative performances.”

- We removed the GENIE3 (non-TF version) from Figure 3B as it was confusing and did not correspond to a common GN inference approach.

3. There are other foundation models built based on large single-cell data, such as scFoundation and biolord. The authors should include them in all the experiments.

Reply: We agree with the reviewer that many new foundation models have recently emerged and that it would be very relevant to benchmark them. However, most of these foundation models are not designed for gene network inference.

In our case, for example, we had to rewrite part of the inference pipeline of scGPT to make it generate networks from scRNAseq data (see our GitHub fork: <https://github.com/jkobject/scGPT>).

While scGPT and Geneformer mentioned inferring gene networks, scFoundation doesn't. Indeed, by using their xtrimogene architecture, only the decoder version of the model can see the full gene-gene matrix. Moreover, the decoder uses the Performer attention method, which never instantiates an attention matrix and thus cannot create a gene network.

Furthermore, although this is a potential way to bypass context length issues plaguing transformers, it has not been used in language and protein modeling due to poor performance (<https://arxiv.org/pdf/2011.04006>).

Making scFoundation output a gene-gene matrix in the same way that scGPT and scPRINT do would be a complex endeavor. It would likely need rewriting the inference pipeline without certainty that it would work since its training was not done with regular attention.

GeneFormer, however, uses the Bert model, and although its inference is more costly than scGPT's, recent updates have made it easier to use (<https://www.biorxiv.org/content/10.1101/2024.08.16.608180v1>). We have used the new Geneformer version, and we have implemented a gene network inference function from it (see our GitHub: <https://github.com/cantinilab/scPRINT/tree/dev/tools>). We show that Geneformer doesn't perform well on most tasks.

- We ran Geneformer and benchmarked it on all our gene network tasks.
- We ran Geneformer and benchmarked it on our cell embedding task.
- We have modified sections 2, 3, and 4 to mention Geneformer's results.
- We now introduce Geneformer in our results section 1.

While scFoundation cannot be used for gene network inference, we still investigate if it could be used for our other benchmarks. While mentioning denoising, we found out that scFoundation doesn't actually output gene expression values but rather gene

embeddings. Indeed in their paper, the denoising benchmark is based on improvement in cell embedding. We thus cannot use it in our benchmark that requires expression profiles. However we were able to use scFoundation for cell embedding and compared it to scPRINT, finding that it performs similarly to scPRINT on this task.

- We have benchmarked scFoundation on our cell embedding task
- We have modified section 4 to mention scFoundation's results

After investigating Biolord, we noticed that it mentioned the idea of a disentangled latent space, which is similar to our notion of a deconvolved embedding for batch effect correction. However, it seems that gene network inference is not one of its features. In addition, it is based on an autoencoder and does not fit our definition of a foundation model. Indeed, despite being able to perform multiple tasks like batch-effect correction and cell-type prediction, the model needs to be retrained on each new dataset. Indeed it doesn't provide pre-trained model weights.

At the same time, many similar VAE models run by the openproblem community are shown in our batch-correction and cell-type prediction benchmarks. Even more methods are available on the openproblem website: <https://openproblems.bio/>. These resources can be used to get an overview of VAE performances on these tasks. In addition, regarding gene network inference, we now added DeepSEM which is a VAE and gives an overview of these methods on these tasks.

- We now cite Biolord in our work, and in agreement with the naming in Biolord's paper, renamed our deconvolved embedding to disentangled embedding.

Reviewer #2:

1. When validating on the Omnipath network, the authors mention that scPRINT-omni "carefully split the dataset into train/test and select, using 50% of the ground truth on the first cell type of each dataset," which suggests a supervised approach to selecting attention heads. However, the authors do not describe how GENIE3 and scGPT handle this. Based on my understanding of GENIE3 and scGPT, they likely use an unsupervised approach to construct gene networks. This could result in an unfair comparison.

Reply: We have now clarified when the different outputs of scPRINT were used and what they were following reviewer 1's comment (see reviewer 1 Question 1).

We also agree with the reviewer and now more clearly explain that our head selection mechanism uses some part of the ground truth while GENIE3's and scGPT's do not.

While we always provide a fully unbiased/unsupervised approach, taking the average over all heads (called simply *scPRINT*), we consider our head selection method to be a novel approach, inspired by the ESM2 paper, which allows a user to refine *scPRINT*'s gene networks according to some prior knowledge. This head selection can be considered as a weak prior on the network since it only selects which of the 128 attention heads is averaged to generate the gene network.

We can also see that the performance of the unsupervised networks, *scPRINT*, is often similar to or better than *scGPT* and *GENIE3*, independently of head selection.

- In Figure 2B, *scPRINT* outperforms *scGPT* (EPR) and *GENIE3* (AUPRC).

- In 2C, *scPRINT* outperforms *scGPT* and *GENIE3*.
- In 3B, *scPRINT* outperforms *GENIE3* and *scGPT* on average in both EPR and AUPRC.
- In 3C, *scPRINT* outperforms *scGPT*.

It shows that *scPRINT* is overall better at inferring gene networks compared to *scGPT* and *GENIE3* even without head selection.

- We have updated paragraphs 5 and 7 of result section 2.

2. The manuscript defines the gold standard for gene networks differently, such as the Omnipath literature-based ground truth and Mcalla et al (ref 24). Many metrics, like AUPR and EPR, heavily depend on the accuracy of the gold standard. When the gold standard is inaccurate, these metric comparisons become meaningless. The authors might consider incorporating reliable simulated datasets, such as BoolODE from BEELINE, for unbiased comparison.

Reply: We agree with the reviewer that the ground truths we use are imperfect. However, simulated data also has issues

(<https://genomebiology.biomedcentral.com/articles/10.1186/s13059-023-02904-1>):

First, these networks are small and often contain only around a dozen genes. Sergio, a more recent approach, has made the tradeoff of simplifying its ODE to increase the number of genes and network size it can simulate. It is also a claim from the paper that adding more genes and their post-processing steps makes the simulated data closer to real expression data.

But since the simulated expression profile is very different from the true biological expression profile, models pre-trained on the true expression profile, such as *scPRINT*, *Geneformer*, and *scGPT*, will be at a strong disadvantage to models that are entirely unbiased and fully trained on Sergio / *boolODE* data, like *GENIE3* and *PPCOR*.

Finally, a model's ability to infer *boolODE* or Sergio's network seems to us more relevant to answer questions about the abilities of transformers and neural networks to invert ODEs.

As it is still interesting to view the results, we have added a small benchmark on Sergio-simulated data in our supplementary tables.

- We have added a supplementary table with benchmarks on a Sergio-generated network.
- We have modified the last part of our results section 2 to mention the Sergio results.
- We have added a new section on our methods section about boolODE, Sergio, and issues with such methods.

3. The sentence "the Omnipath literature-based ground truth does not reflect the topology of a biological network" is unclear. Why doesn't it reflect network topology?

Reply: We agree with the reviewer that this sentence is unclear. It is hard to say what a biological network should look like. Still, it is sensible that a network made from literature-based information will be at least skewed towards highly studied genes and interactions that can be easily validated.

<https://www.ncbi.nlm.nih.gov/pmc/articles/PMC4523822/>

<https://www.nature.com/articles/s41598-018-19333-x>

Given that this was not the main information we wanted to deliver, we removed the sentence to avoid confusion.

- We changed the sentence in the first paragraph of the 3rd results section: "Although we have shown that our networks represent meaningful biology, the Omnipath ground truth is literature-based and not cell type-specific. "

4. The primary contribution of this paper is the construction of cell-specific gene networks. Previous studies have referred to these cell-specific gene networks as dynamic networks (see Citation1 and 2), often utilizing them in dynamic processes such as cell development and differentiation. However, this paper does not address the analysis of changes in gene networks during processes like cell development and differentiation, which I believe is a significant shortcoming.

[1] Nature Communications, 2023, 14(1): 8459.

[2] Nature Methods, 2023, 20(9): 1368-1378.

Reply: Indeed, this work focused on what could be defined as steady-state gene networks. Many cell types and tissues can hardly be studied as dynamic processes. Moreover, most datasets do not contain the necessary information to infer trajectories, while those that can, sometimes generate aberrant trajectories.

Inferring dynamical networks is outside the scope of this work and our approach. However, we did consider in our conclusion that this is a shortcoming of many gene network inference techniques, including ours, and of the current sequencing technologies available.

- In the first paragraph of the main section, we discuss the difference between these two modalities of gene network inference.
- While we had cited dictys already, we now also cite CEFCON in the main section.
- In the limitations and remaining work of the discussion section of our manuscript, we add the lack of “time” and “space” features.

5. Large foundation models are not the only solution for constructing gene networks based on single-cell RNA-seq. Recently, there have been numerous methods specifically designed for gene network construction using scRNA-seq that do not rely on pre-training[1],[2],[3]. The authors need to further demonstrate that scPRINT, which utilizes large model pre-training, is superior to these methods tailored to specific datasets. Comparing with GENIE3 alone is insufficient, as GENIE3 is a method from over a decade ago and was not specifically developed for scRNA-seq data.

[1] Nature Computational Science, 2021, 1(7): 491-501.

[2] Nucleic Acids Research, 2023, 51(4): e20-e20.

[3] Briefings in Bioinformatics, 2023, 24(5): bbad326.

Reply: We agree with the Reviewer that many more methods that are not foundation models exist for gene network inference.

However, many have been developed for specific data types. We find three main categories: temporal, steady-state, and knock-out-based. Our work has focused on the more challenging and general class of steady-state gene network inference techniques. The BEELINE paper (<https://www.ncbi.nlm.nih.gov/pmc/articles/PMC8138892/>) showed that despite its simplicity, GENIE3 was one of the best-performing methods among the tested ones.

Considering the novel methods proposed by the reviewer, we chose to use DeepSem and benchmark it on all our gene network inference and embedding benchmarks.

Indeed, it appears that Normie is temporal-based and wouldn't work on our benchmarks. It is also TF-gene specific, while only one of our benchmarks is TF-gene specific.

<https://github.com/CSUBioGroup/Normie>

<https://academic.oup.com/bib/article/24/5/bbad326/7274797>

Finally, one of our goals was also to showcase well-maintained and easily usable tools. scGRN contained too little information to know how to run the tool, what the input file format was, or what the output was. Given the lack of information in GitHub, we could not run the example from the README file.

- We chose to use DeepSEM and benchmark it on all our gene network inference
- We also now introduce DeepSEM in our introduction
- We contextualize the different types of gene network inferences in our introduction: "Other methods consider datasets where differentiating cells can be ordered temporally to predict more causal GRNs. While this approach is interesting, temporal ordering is often hard to predict"
- We also now add a paragraph about DeepSEM and the choice of hyperparameters in our methods section
- We also modified sections 2 and 3 to talk about DeepSEM's results

By adding DeepSEM, we realized we did not mention the critical fact of network inference speed. We have added a sentence about this in our manuscript.

- We have updated the last paragraph of results section 1 to mention inference speed.

6. "In contrast, head selection is often more advantageous in larger models with more heads (see Table S4)", Table S4 is missing in the supplementary file.

Reply: we have now corrected this omission in the manuscript, and it is now Table S6

7. In line 342, the authors mention, "While scPRINT does not train on the test dataset itself and makes predictions over hundreds of labels, it still reaches 62% classification accuracy (Figure 4B, Supp Figure S5)." However, the comparison methods chosen were logistic regression and xgboost. Why was scGPT not selected for comparison? 62% classification accuracy is not particularly high, even without fine-tuning.

Reply: scPRINT's approach differs from models such as xgboost or scGPT as its prediction ability is zero-shot. Indeed, making scGPT predict cell type requires adding a classifier on top of it and training it together with its cell embedding, which is often called "fine-tuning". This is the same approach used for scFoundation, and it explains why we didn't add scGPT or scFoundation, as their abilities on these tasks are supervised on the same data instead of being zero-shot.

Only a marker-based approach that doesn't involve having access to the training data would be a fair comparison here. For this reason, we have added CellTypist to the comparison.

In addition, reaching 62% accuracy is nontrivial when using a model that was not trained on the data itself. It also has to predict over 200+ labels, compared to the 10 labels that other models trained on the dataset have to choose from. Indeed, the chance level in this context would be 1 in 200+ without taking into account the possible biases of cell

type that are more or less frequent in the training set. Compared to 1 in 10 for the other models, which are also heavily biased to the proportion of cell types in the test dataset. We have now clarified the nontriviality of this task.

- We have added a more comparable method of marker-based annotation: CellTypist.
- We have modified section 4 to discuss CellTypist's results and the nontriviality of the inference task for scPRINT.
- We have added a part in our method to talk about CellTypist and the choice of hyperparameters.
- We have added a mention about fine-tuning in our methods section.

8. According to the methods section, the embedding for a specific gene expression includes ESM2 embeddings, gene expression embeddings, and gene position embeddings. However, the ESM2 embeddings and gene position embeddings are identical for the same gene. The variation in attention calculation across different cells mainly depends on the gene expression embeddings. Since the embedding for expression is merely an MLP-upscaled version of the log-transformed expression value, I have some doubts about the effectiveness of this representation approach.

Reply: We need to encode some information about the gene ID as transformer models lack positional information of their input. Indeed, Transformers work on a set of input elements and need either an encoding of the elements themselves and/or of the position of their input elements.

scGPT and Geneformer solve this "tokenization" issue by learning embeddings of each gene ID together with the model weights. This adds $\text{model_dim} \times \text{num_genes}$ (e.g., $512 \times 50,000 = 25\text{M}$) more parameters.

Instead, we decided to use the ESM2 embedding of the amino acid of each gene's protein sequence. This embedding first lets the model differentiate amongst the genes it sees. Moreover, it also provides structural and evolutionary similarity information about genes [<https://www.science.org/doi/10.1126/science.ade2574>, <https://www.biorxiv.org/content/10.1101/2022.07.20.500902v1.full.pdf>]. Finally, it allows scPRINT to make inferences on genes it hasn't seen at test time. Adding information about a gene's location in the genome could be seen as redundant since it also encodes information about the gene ID. Our goal is to use gene ID to encode as much information about it as possible, here its AA structure and genomic location.

The variation in attention from two cells will only come from the set of genes expressed and their expression values. We are very much interested in the overall attention patterns the model will learn and not just its variation. the possible connections to be

explored are astronomically large and giving information about the gene location and protein AA structure should help the model prioritize biologically meaningful ones.

To look at the importance of these features to the downstream tasks we have performed an ablation study shown in our supplementary table S3. We show there that losing positional encoding decreases performances slightly, although one cell type prediction benchmark gets better results. It is more striking for the denoising benchmark which drastically loses in performance when using a model that hasn't been trained with positional encoding (medium-noPE). This might be because genes that are close on the DNA often tend to have correlated expression profiles. scPRINT can learn to use this information to improve its ability to denoise expression, better associating genes that are close together on the genome.

We also show that learning the gene ID (medium-nofreeze) instead of using ESM2 doesn't change the results much but might slightly improve denoising ability. However, this is at the expense of not being able to work on unseen genes and a poor embedding of more rarely expressed genes.

Regarding the encoding of each gene's expression value, we need to convert a scalar value to a vector or "tokenize" it, by using an MLP with two layers, we effectively let the model learn the "metric" of transcription expression. Moreover, again, we decrease the number of parameters used compared to scGPT while being able to make predictions on count values unseen during training, such as those of bulk or pseudo-bulk RNAseq.

- In paragraphs 5 and 6 of the first section of the results, we updated the description of the model's input and encoding.
- We have updated the methods section to make this part more straightforward.

9. "Taking as an example the differential hub PAGE4 (see Figure 4C), we see that it is connected to many of the top 15 hub nodes in the BPH network," This should be Figure 5.

Reply: We corrected this mistake in the manuscript

10. In gene network prediction, the characteristic of gene relationships, such as activation or inhibition, is also crucial. The authors should discuss whether scPRINT can potentially infer the characteristic of these gene network relationships.

Reply: We agree with the reviewer that deciphering inhibition from activation is an important and challenging problem. Like most methods, scPRINT doesn't differentiate between the two. In principle, this could be done by either selecting a set of scPRINT heads that correlate with inhibition if any exist or querying scPRINT's networks by computing a categorical jacobian

[<https://www.biorxiv.org/content/10.1101/2024.01.30.577970v1>]. Still, with the scarcity of inhibitory relationship data, we did not test it in our work. However, because of the scarcity of ground truth information, most benchmarks have not validated inhibition scores.

Indeed, for omnipath, only a few thousand inhibitory relationships were available (less than 1% of the total), which we deemed insufficient.

For McCalla, which is based on overlaps of multiple sources, the inhibition/activation relationship was not given in the database list of interactions. Moreover, the ground truth's small size was already giving issues for benchmarking, and decreasing it even more might lead to too much variance. For GWPS, only the q -value of the differential expression tests was made available and not the fold changes (which could be used to inform the inhibition/activation relationship). Recomputing this missing information for 20,000 genes * 8500 knock-outs would be exceptionally computationally costly.

We have now added a discussion on activation and inhibition in our methods section. In summary,

- In the 8th paragraph of the first section of the results, we mention that while we don't consider the sign of our ground truth network, our head selection method could learn to predict it given enough supervision information.
- We mention it as a limit of our work in the 7th paragraph of the discussion.

Reviewer #3:

1. It would be helpful to have a guideline on the general recommendations as to which type of scPRINT model to use in practice. The figures in 2B suggest the use of scPRINT-mean or scPRINT-omni, while I believe in 3B, a different version of scPRINT(-self) is more appropriate. In particular, in 3B, it seems that only scPRINT-self can "outperform" random guessing. The performance appears better in Figure 3C, but we see a more varied result there, with Genie 3 being competitive. How do the scPRINT models used in Figures 2/3 compare to the scPRINT models in Fig. 4?

Reply: We agree with the reviewer and have modified the manuscript to clarify the different network outputs of scPRINT and their usage.

We want to clarify that the same instance of scPRINT was used across our manuscript. The different names used in the gene network inference tasks (Figures 2, 3) refer to how we aggregate the heads of the model to form the final gene network. We clarified it as part of our answer to Reviewer 1 Questions 1 and 2.

We also realized that the figure legend did not describe the scPRINT network names used in Figure 3.

scPRINT can produce two types of networks:

- One uses the mean of all attention heads (now simply called scPRINT), which is not specific to any interaction type (we call it a gene-gene network).
- One uses the mean of selected attention heads using some ground truth information (now referred to as scPRINT([ground truth name]'s heads), e.g., scPRINT(omnipath's heads), when using omnipath for selecting the heads). This version might be more specific to the interaction types present in the ground truth.

Both can be subsetted to TF-gene networks (GRNs) by removing non-TF ->gene links. This is the same approach we have now used for DeepSEM and Geneformer.

For GENIE3, however, we have to run it twice. Once using all genes as predictors and once using only TFs.

We now modified the text to address these points:

- We have renamed scPRINT-self to scPRINT(Han et. al.'s head): heads selected with the Mc Calla et al. ground truth, scPRINT(gwps' heads): heads selected with the gwps ground truth, and scPRINT(omnipath's heads): heads selected with the omnipath ground truth and updated the figure legends.
- We have updated paragraphs 5 and 7 of result section 2.
- We have changed the last part of the first section of the results.
- We have changed the end of the first section of the results.
- "The attention heads are either all aggregated by averaging or can be selected to better reflect connections of interest (Figure 1C). This is done using the average of the heads most correlated with literature or perturbation-based ground truth networks. Finally, while we do not assess scPRINT's ability to model inhibition due to the scarcity of such annotations, we leave open the possibility of using our head selection technique for such a task."
- We improved our explanation of the output of scPRINT across the different tests:
- "Of note, due to the small number of genes assessed in the ground truth, we do not add the genome-wide network versions here. ", and
"Moreover, only the TF version of GENIE3 and the TF-gene subsets of the other method's networks are used since the ground truth only contains TF-gene connections." in section 3 of the results, and
- "For scPRINT, we generate three network versions: one simply called scPRINT, based on the average of all heads in the model. scPRINT (omnipath's heads), based on the average of heads selected with our abovementioned head selection method inspired by ESM2, and scPRINT (genome), which is like the scPRINT network but uses our method to generate genome-wide networks (see Methods) instead of using the 5000 most differentially expressed genes." in section 2 of the results.

- We have added a supplementary figure about TF-gene, only part of the GWPS
- we make it clear that both are the same for gene networks and gene regulatory networks in paragraph 9 of section 2 of the results
- We removed the GENIE3 (non-TF version) for Figure 3B as it was confusing and did not correspond to a common GN inference approach.

Finally, we have added guidelines on which gene network to use in practice in the discussions: "Given the discrepancy amongst ground truth networks, we advise users to consider using all-head averaging and to only revert to head selection when some high-confidence interactions are available. Indeed, general collections like Omnipath did not improve performance in most of our tests."

2. Gene embedding that aggregates multiple lines of evidence is interesting. However, it's unclear how different types of embedding results play a role in downstream analysis. Especially how AA sequence information and genomic positional embedding modulate gene expression embedding patterns are worth investigating.

Reply: We agree that more work remains to be done in the field of transformers applied to biological data, in general, to explore the different kinds of input encoding and benchmark them. However, the model needs both information about the gene ID and gene expression to make predictions.

Indeed, transformer models lack positional information of their input, they need the information of the gene "id" or "name" for which we provide the expression. Indeed, Transformers work on a set of input elements and need either an encoding of the element themselves and/or of the position of their input elements (<https://arxiv.org/pdf/1706.03762>).

scGPT and Geneformer solve this "tokenization" issue by learning embeddings of each gene ID together with the model weights. This adds $\text{model_dim} \times \text{num_genes}$ (e.g., $512 \times 50,000 = 25\text{M}$) more parameters. Instead, we used the ESM2 embedding of the amino acid of each gene's protein sequence

(<https://www.biorxiv.org/content/10.1101/2022.07.20.500902v1.full.pdf>).

This embedding first lets the model differentiate amongst the genes it sees.

Moreover, it also provides structural and evolutionary similarity information about genes (<https://www.science.org/doi/10.1126/science.ade2574>). Finally, it allows scPRINT to make inferences on genes it hasn't seen at test time. This part does not encode expression information but just the gene ID.

Encoding the expression values of each gene requires converting a scalar value to a vector or "tokenizing" it. Geneformer used positional encoding, a function often used to encode the position of words in a text (<https://arxiv.org/pdf/1706.03762>). Similarly to

gene name token, scGPT learned an embedding for different ranges of expression values, binning them to remove sampling noise.

Both approaches apply a specific prior to the metric that defines expression.

Geneformer puts an equal distance between each value: an expression of 2 is as far from an expression of 1 as 123 is from 122. Meanwhile, scGPT says that 1-2-3 and 4-5-6 are the same, whereas 4-5-6 are different from 1-2-3 by some amount learned by the model.

We sum this embedding with the gene ID embedding and the gene location embedding.

AA sequence and genomic location do not change the “encoding” of the expression, but there might be a possibility to create a collision between the different information encoded by each. Which may cause some information to be lost.

However, adding multiple representations as input to a transformer is common in most domains (e.g., positional encoding and gene ID in geneformer, position encoding and word ID in LLMs, gene expression ID and gene name ID in scGPT).

The main idea is that since one of the two representations is learned (here, the expression), the model will make it robust to the potential perturbations introduced by the other one. This is always possible and expected, given a sufficiently high dimension of the embedding space.

AA sequence and genomic location, however, will provide information to the model, modulating its downstream prediction of gene expression. We have asked the question of their impact on the model’s predictions with an ablation study seen in supplementary table S3. We show there that losing positional encoding decreases performances slightly, although one cell type prediction benchmark gets better results. It is more striking for the denoising benchmark which drastically loses in performance when using a model that hasn’t been trained with positional encoding (medium-noPE). This might be because genes that are close on the DNA often tend to have correlated expression profiles. scPRINT can learn to use this information to improve its ability to denoise expression, better associating genes that are close together on the genome.

We also show that learning the gene ID (medium-nofreeze) instead of using ESM2 doesn’t change the results much but might slightly improve denoising ability. However, this is at the expense of not being able to work on unseen genes and a poor embedding of more rarely expressed genes.

- We explain more clearly the way we encode scPRINT’s input in section 1 of the results.
- We have added a part to the methods about the ablation study.

3. The selection of attention heads for scPRINT could be explained more clearly, together with its implications on interpretability and further downstream usage.

Reply: We agree with the reviewer and have improved and expanded this part of our manuscript thanks also to questions and remarks from Reviewer 1 (questions 1 and 2), Reviewer 2 (question 1), and Reviewer 3 (question 1). We addressed concerns about the attention head selection mechanism by renaming the heads, updating our introduction of head selection, and discussing which to choose as well as their limitations. In addition, we have updated some additional elements based on this comment, highlighting our recommendation on which method to choose and a discussion on interpretability:

- We have added an explanation of which scPRINT version to use in which condition in the second paragraph of the discussion section
- We now discuss further the question of interpretability in the methods section about head selection
- In the 7th paragraph of the discussion, we discuss the limitations of scPRINT in distinguishing the types of interactions it infers (due to the scarcity of ground truth).

4. What type of pretraining tasks were performed? Sure, we can speculate based on previous work (and figures), but it is not fully stated in the texts. Were these different tasks combined or separate and sequential?

Reply: The 3 pre-training tasks are denoising, classification, and bottleneck learning. Each loss is added up and trained conjointly. We improved a large part of section 1 of our results to present more clearly the pretraining tasks used for scPRINT. We have also undertaken a lot of efforts to make scPRINT as reproducible as possible. Our pretraining tasks, model, datasets, weights, benchmarks, dataloaders, and preprocessing methods are made available and packaged. It has been successfully used by multiple researchers already. The pre-training details are also mentioned in more detail in the methods section called "Pretraining".

- We have updated paragraphs 2,3,4, and 5 of the first section of the results to explain much more clearly our pretraining tasks and their relationship to gene network inference:
"scPRINT's novel pretraining is composed of three tasks which loss are added and optimized together: a denoising task, a bottleneck learning task, and a label prediction task. Their objective is to let scPRINT learn to represent meaningful gene connections while also endowing it a breadth of zero-shot prediction abilities... "

5. Please consider using different fonts in the main figures. Some parts are not clearly legible.

Reply: While we were not sure which element of the figure was referred to, we have increased the font size of the smallest font text in all the main figures. We have also ensured that the same fonts were used across all our main figures. Given more details and examples about which elements to change, we remain open to updating our figures further.

- We have increased the figure and graphics minimal font size.
- We have increased the font size of the umap labels. We have taken this chance to also update the umap by including even more cells (1M->2.5M and more cell type labels)
- We have increased the font size of the differentially expressed gene names in Figure 5
- We have changed some mismatched fonts. The figures now contain 2 Fonts (Arial & Scribe) instead of 5.

6. Why 2,200 genes rather than 20k genes? Other methods take advantage of full coding genes (e.g., Hao .. Song, Nature Methods, 2024). I think embedding could handle a flexible number of genes even though many of them were cropped in cell x gene DB.

Reply: The only method using all 20k genes we know of is scFoundation from Song et al. They can do it by applying 2 strategies:

1. their xTrimoGene architecture that only uses expressed genes, with variable batch size during training, incurring an important memory cost on most hardware as it is constrained by the largest possible batch size.
2. And the use of a Performer for the attention mechanism, which approximates attention. Indeed the bottleneck of the transformer is the attention mechanism which grows in $O(n^2)$ with the context length n .

Performer is an often-cited method and part of the literature on attention approximation. However, most state-of-the-art transformer models do not use attention approximation as they are known to lead to worse performance (<https://arxiv.org/pdf/2011.04006>). Moreover, in cellxgene, >80% of the cells have less than 2200 genes being measured. This means that only less than 1/5th of the cells would benefit from a larger context. In other words, for scFoundation, most of the memory and compute power is likely lost on tokens that are almost always zeros due to dropout.

We agree that work involving novel architecture is important to large cell models, although in-depth independent reviews have yet to be performed on a broad range of tasks

Finally, our manuscript shows that although scPRINT trains on “only” 2200 genes per cell, it can perform well on larger contexts at inference time. In Figures 2 B, C, and D, we use 5,000 most variable genes as input and up to 20,000 genes in our *scPRINT(genome)* network.

- We have added a note on context length in the methods
- We have made updates to the 12th paragraph of section 1 of the results “scPRINT is pretrained using 2,200 randomly selected expressed genes in a cell profile. If a cell doesn’t have enough expressed genes, the list is padded with randomly selected unexpressed genes. While not genome-wide, this captures all of the expressed genes in more than 80% of the cells in the cellxgene database while still retaining a full attention computation (see Methods). Moreover, we show that scPRINT can make predictions on much larger sequences of genes at inference time.”

7. Comparison with other ML models needs more clarity. What the training and testing data sets were and how hyperparameters were tuned could certainly change the performance.

Reply: We have updated the text regarding the description of the training and testing of scGPT and GENIE3 and their hyperparameters.

We have also made a clearer description of the training and testing of the non-gene-network-inference tasks.

For batch correction and cell type prediction, we say that the parameters were defined by the openproblems benchmark and available on their paper. We have added the specific hyperparameters used for the denoising task.

- In the Results section 4, we have made it clearer that the results of cell type prediction and batch correction come from openproblems, and we mentioned that hyperparameters are set by them
- We have mentioned hyperparameters used in the denoising and classification tasks in their methods section.
- We have mentioned any hyperparameters used for genie3 and scGPT outside of the default ones in their associated methods section.

8. Non-transformer models are expected to perform worse, but it would be worth confirming that these methods also used full-scale data (with random sampling).

Reply: We understand it could be an issue if the gene network inference methods were done on different sets of randomly sampled cells. We now clarify that all methods use the same set of cells across all models benchmarked.

There may also be a question about the genes used across models. Again, we have clarified that each method used the same set of genes.

We want to explain also that although the model is pretrained with a random sampling of genes, it does not have to use random sampling during inference, and in most of our experiments, it doesn't.

Finally, training methodologies vary greatly across models. scGPT is trained similarly to scPRINT, while GENIE3 is not trained per se.

For the benchmarks of section 4 of the results, the same data is given to each model, but the preprocessing is method-specific. While some use PCA, other expects most variable genes or other transformations. This decision is defined by open problems based on the common ways to run each tool.

- We have added, for each non-gene-network-inference task, that the same set of genes and cells were used
- We have added the mention that the same set of genes and cells were used across the models for all gene network inference tasks

Reply to Reviewer 1:

I first want to thank the authors for responding to my question in detail. But I still have some concerns:

1. Regarding the interaction type that scPRINT can predict. Can you give some examples of how protein-protein interactions and protein-TF interactions can be predicted if we only input gene expression to the scPRINT? One gene generates multiple proteins. It is challenging to predict those interactions using only gene expressions.

Reply: Indeed, the predicted interactions are not at the protein level. From gene expression and some additional priors, scPRINT predicts interactions between genes (i.e., gene networks). These predicted gene interactions are undetermined and could reflect true biological interactions between the protein products of these genes or some other form of interactions, e.g., the transcripts or protein product of one gene to the transcript or some regulatory element of the other gene. scPRINT doesn't make this distinction per se, a limitation that all other gene network inference methods have.

To evaluate the quality of the inferred networks, we look at what is known about the interactions of these genes. It is common for gene network benchmarks to look, for example, at PPI networks where proteins are mapped back to their corresponding genes. This is what we do in our benchmark as well. We sometimes refer to our "predicted TF-gene networks" because we restrict the predicted gene-gene interactions only to TF → genes interactions. Here again, "TF" means genes encoding transcription factor proteins. These predictions are benchmarked with ground truths from experiments assessing the binding of proteins (TFs) to the regulatory region of the genes.

We have added a supplementary section in the method to explain the type of molecular interactions we output and the ones we use as our ground truths:

"We generate the Omnipath network using all the interactions from the Omnipath Python package, excluding small molecules, lncRNAs, and any element without a unique HGNC symbol. We then transform it into a directed binary network of source to target. These interactions are extracted from the literature and represent mainly TF to gene connections as well as many protein-protein interaction connections and small amount of other connection known from the literature like RNA-RNA interactions, protein-RNA interactions and more. All interactions are mapped back to their gene IDs, generating a gene-gene network encompassing the various interactions the genes and their molecular products can have."

"All methods presented here generate networks of their input data. Given gene-level expression data, they will generate gene-networks. Without additional information, no

method can distinguish the type of molecular interactions that underpin their predicted network edges. “

2. In scFoundation paper, it shows that it can predict TF-gene interaction in the last section. You should compare scPRINT with it.

Reply: Indeed, scFoundation now has a small two-paragraph results section about “Inferring gene modules and gene regulation networks,” which it did not have in its initial preprint: <https://www.biorxiv.org/content/10.1101/2023.05.29.542705v1.full>. In this new section, they use scFoundation’s output gene embeddings to generate gene modules (clusters of genes without links). In addition, they have a GRN inference part, which is, however, based on applying scenic to the output gene embeddings. This means that scFoundation doesn’t generate gene networks by itself.

Indeed, Hao et al. perform clustering of gene embedding of the last layer of the model to extract “gene modules” and benchmark them using enrichment for cell type markers. Because of the learning task of scFoundation and similar foundation models, the last layer’s embeddings mostly contain information about the gene’s expression since it is what will only be predicted from that embedding. Thus, it makes sense that the “differential gene module” analysis performed by Hao et al. will show differential expression between cell types and, thus, cell type markers.

For the Scenic analysis, Scenic is usually run with GENIE3 as the first inference part of the network and then subsetting to the TF with known motifs and to TF-gene connections where the TF motifs lie near a gene transcription start site (RcisTarget). This is not the way it has been applied here. Similarity is computed from the output embedding of scFoundation on 1000 of the most variable genes in a dataset. From it, the 1000 TF-gene connections with the highest similarity are kept (they will mostly relate genes with similar expression values). This is then used directly as input to RcisTarget. This means that the gene network is mostly driven by RcisTarget. This inductive bias could be applied to all gene networks we have analyzed in our manuscript is something we elected not to use.

Because of its xtrimogene architecture, extracting gene networks from scFoundation in the same way we have done with Geneformer, scGPT, and scPRINT is not straightforward. Indeed, scFoundation doesn’t focus on this application. Other approaches, such as using perturbation predictions as a proxy across all models, might be possible, but they fall outside the scope of this work.

We have made an addition to our methods section to explain further why scFoundation could not be used despite its mention of gene networks:

“We could not validate scFoundation on our Gene network inference benchmark as

extracting a network from the attention matrices was much more complex due to the xtrimogene architecture. scFoundation mentions the generation of gene modules using clustering of its output gene embeddings. It also mentions the interference of gene networks. However, it is achieved using RcisTarget[<https://www.nature.com/articles/nmeth.4463>], a prior gene network based on motif analysis. This approach is not comparable to the gene networks generated by scPrint, Geneformer, and scGPT. Indeed, RcisTarget could be applied to every model we have benchmarked and would prevent us from doing an unbiased benchmark. Neither our approach nor Hao et al.'s could extract gene networks directly from scFoundation. It is being left to further investigations.”

We remain available to perform any additional analysis required by the reviewer.